# Real-time Personalized Federated Continuous Learning via Generative Replay

## Abstract

Recently, Federated Continuous Learning (FCL) has gained attention for simulating real-world dynamic problems, with catastrophic forgetting as its core challenge. While generative replay is widely used in FCL methods to mitigate this issue, higher cross-client data heterogeneity necessitates excessive FL rounds per task for convergence, thereby conflicting with clients' demand for immediate responses. To address this, we focus on real-time FCL, where incremental data arrives in small batches per FL round and is only accessible at that FL round, causing global data heterogeneity to vary across FL rounds, and propose pFedGRP, which includes two key components: Firstly, a flexible generative replay architecture that decouples the generator by category to mitigate inter-class catastrophic forgetting, combines with the task model to reduce redundant updates and improve generation quality, and adaptively adjusts client-specific local generation scales. Next, a personalized FCL framework via generative replay that optimizes aggregation weights on server-side for real-time model personalization, and transfers personalized knowledge to an extra average global model on client-side for catastrophic forgetting mitigation. Experiments show pFedGRP outperforms other FCL methods via generative replay, with both superior performance and lower regret.

## 1 Introduction

Federated Learning (FL)(McMahan et al., 2017) is an emerging privacy-preserving distributed machine learning (ML) framework. Personalized FL (pFL)(Tan et al., 2023; Sabah et al., 2024), which balances client utility with collaboration, has gained notable attention. However, the static single-task framework of both FL and pFL lacks real-world practicality: in practice, clients receive data randomly and incrementally, causing dynamic shifts in intra/inter-client data heterogeneity across FL rounds(Li et al., 2020a; Kairouz et al., 2021). Additionally, regulations(Voigt & Bussche, 2017; Vizitiu et al., 2019; Balogun, 2025) constraints further limit raw data retention. For example, health agencies in different regions can utilize FL to conduct research on COVID-19(Yang et al., 2020; Dayan et al., 2021), but the virus's rapid mutation rate leads to significant variations in regional data distribution and trends, while privacy regulations(Voigt & Bussche, 2017; Balogun, 2025) limit the retention period of raw medical data. In this case, FL and pFL are prone to catastrophic forgetting(Kirkpatrick et al., 2016; Kemker et al., 2018)—a phenomenon where models forget prior task knowledge when learning new tasks without access to original training data.

In centralized ML, continuous learning (CL) works(Wang et al., 2024a; 2025) mitigate catastrophic forgetting through approaches such as rehearsal(Lee et al., 2024; Tong et al., 2025), generative replay(Wang et al., 2024b; Chen et al., 2025), regularization(Lin et al., 2024b; Bian et al., 2024), and parameter isolation(Lin et al., 2024a; Hu et al., 2024), among others. But in Federated Continuous Learning (FCL)(Yang et al., 2024), distributed scenarios raise task ID prediction difficulty for regularization and parameter isolation approaches, while privacy regulations constraints hinder rehearsal approaches, promoting generative replay, with most FCL methods adopting a global generator to mitigate catastrophic forgetting. However, these FCL methods mainly focus on offline CL, with each task spanning dozens to hundreds of FL communication rounds (i.e., FL rounds) for model's global aggregation and fine-tuning to achieve convergence. In contrast, real-world applications resemble online CL (OCL)(Lange et al., 2022; Gunasekara et al., 2023), with client data arriving per FL round in mini-batches and accessible solely within that FL round, causing less label/feature distribution overlap among many clients per FL round. We call it real-time FCL to distinguish it from

online CL. In real-time FCL, FCL methods with single global generator and task model will struggle to address dynamic data distribution shifts due to convergence lag (Zhu et al., 2021; Cao et al., 2023), degrading performance for clients' real-time need and weakening catastrophic forgetting mitigation. Therefore, improving FCL for real-time FL aggregation through generative replay is crucial.

In this work, we integrate generative replay into pFL for real-time FL aggregation while mitigating catastrophic forgetting. Firstly, we propose a flexible generative replay architecture: since model's training resource needs scales with data volume under a fixed batch size(Gui et al., 2023), to avert cross-category catastrophic forgetting, we replace existing FCL methods' single larger global generator with multiple smaller class-wise sub-generators, utilizing compact existing generative models. Moreover, we use the latest task model to identify sub-generators' feature drift, reducing redundant updates thus speeding generator training. Subsequently, we use the latest task model to enhance sub-generator's output quality, better mitigating task model's catastrophic forgetting on client-side while enabling server-side real-time personalized FL aggregation. Furthermore, to handle diverse task paths across clients, we adaptively adjust local generation scales to reducing replay errors.

Based on upon, we propose pFedGRP, a real-time FCL via generative replay. For real-time personalized FL aggregation, servers sync sub-generator caches through clients' updated sub-generators, then replay client-specific distributions to optimize personalized aggregation weights per FL round. To align the global feature space for faster convergence, we propose using dual global task models: clients initialize local task model with the global average one per FL round, then inject personalized knowledge through replay data during local training to mitigate catastrophic forgetting.

Our main contributions can be summarized as follows:

1.We introduce a novel FCL framework for real-time FCL, tackling slow convergence in generative replay-based FCL methods that requiring multiple FL rounds per task.

2.We propose a flexible generative replay architecture with class-specific sub-generators that enhanced by task model to support task model's real-time personalized FL aggregation.

3.Based on above, we propose pFedGRP, a real-time FCL method via generative replay, achieves task model's global real-time personalized aggregation and local catastrophic forgetting mitigation.

4.Leveraging benchmark datasets, we propose scenario construction schemes and proper metrics for real-time FCL, validating our approach across diverse datasets and dynamic data scenarios.

## 2 RELATED WORK

**Federated Learning and Personalized Federated Learning**: Federated Learning (FL)(McMahan et al., 2017) enables distributed ML without raw data transmission, with its core challenge being constructing the global model that performs well across clients with heterogeneous data distributions. One approach is to enhance knowledge transfer within the feature space of the single global model, including methods such as parameter difference constraints(Li et al., 2020b), statistical information sharing(Li et al., 2021b), dynamic regularization(Acar et al., 2021), hidden space alignment(Yoon et al., 2021b), dual label correction(Wu et al., 2023), and global aggregation fine-tuning(Li et al., 2023), among others. Another approach is to customize a global model for each client by adjusting collaboration with others, known as personalized federated learning (pFL), including methods such as model distance estimation(Li et al., 2021a), model representation decoupling(Collins et al., 2021), model hierarchical sharing(Arivazhagan et al., 2019), personalized knowledge transfer(Zhang et al., 2021), personalized distribution mixing(Marfoq et al., 2021), and personalized collaboration graph(Ye et al., 2023), among others. However, current FL and pFL methods are designed for static local data (a single task) and need multiple FL rounds to converge. In FCL scenarios, when learning new task, these methods struggle to mitigate catastrophic forgetting caused by the inability to access old task data, resulting in degraded performance across all tasks.

**Federated Continue Learning**: Federated Continuous Learning (FCL) extends CL to FL scenarios. In this paradigm, each client's local dataset remains static for each task that spans multiple FL rounds, while federated collaboration are employed to retain cross-task knowledge and mitigate catastrophic forgetting. Current FCL works mainly focuses on: Task parameter isolation methods(Arivazhagan et al., 2019; Yoon et al., 2021a; Luopan et al., 2023; Wang et al., 2024c) train task-specific modules but require explicit task ID provision during training and inference. The orthogonal

model update methods(Shoham et al., 2019; Bakman et al., 2024; Salami et al., 2025; Zhang et al., 2025) compute task residual matrix for parameter update, but small-batch's high sample variance in real-time FCL severely challenges this computation. The sample replay (i.e., rehearsal) methods(Li et al., 2024; Nori et al., 2025; Serra & Buettner, 2025) preserve local core sets through task model, highly rely on task model performance and may violate data privacy laws. There generative replay methods mainly adopt two strategies: local training methods(Qi et al., 2023; Wuerkaixi et al., 2024; Serra & Buettner, 2025) update global generator on client-side per task through multiple FL aggregations, global training methods(Zhang et al., 2023; Babakniya et al., 2023; Tran et al., 2024) update global generator on server-side with the converged global task model after per task's completion, both strategies require a large-scale generator to fit the complex global feature distributions of all tasks across multiple clients, causing computational and scalability challenges in training and inference. In short, most FCL methods need multiply FL rounds for global model convergence per task. In real-time FCL, poor convergence degrades model performance(Zhu et al., 2021), failing to meet clients' demands per FL round.

## 3 PRELIMINARY

**Symbol Definitions**:

For the model, we denote the task model used to solve practical problems as $\boldsymbol{\omega}$, the generator for all categories as $G$, the smaller sub-generator for a category $c$ as $G_c$. In the $t$-th FL round, $\boldsymbol{\omega}_i^t$, $G_i^t$ and $G_{i,c}^t$ represent the local models updated on the $i$-th client $C_i$, while $\boldsymbol{\omega}_g^t$ and $\boldsymbol{\omega}_{g,i}^t$ refer to the global task model and client $C_i$'s personalized global task model aggregated on server, respectively.

For the local dataset, we denote the local dataset of client $C_i$ in the $t$-th FL round as $D_i^t = \{(x_j, y_j)|\forall j \in [m_i^t]\}$, with data distribution $\mathcal{P}_i^t = P(\mathcal{X}_i^t, \mathcal{Y}_i^t)$ defined as the joint distribution over its label space $\mathcal{Y}_i^t$ and feature space $\mathcal{X}_i^t$. The local dataset $D_i^t$ consists of $m_i^t$ sample pairs, with each $x_j \in \mathcal{X}_i^t$ as the input and $y_j \in \mathcal{Y}_i^t$ the corresponding label. Moreover, let $Y_i^t = \{m_{i,c}^t|\forall c \in \mathcal{Y}_i^t\}$ denote the data quantity vector of $D_i^t$, with $m_{i,c}^t$ as the sample count of category $c \in \mathcal{Y}_i^t$.

For the synthetic dataset, we denote client $C_i$'s local label space as $\mathcal{Y}_i = \cup_{t=1}^T \mathcal{Y}_i^t$ (across all $T$ FL rounds), and the synthetic dataset created by generator $G_i^t$ as $\tilde{D}_i^t$("$\sim$" denotes "synthesis"). Let $\tilde{Y}_i^t = \{\tilde{m}_{i,c}^t|\forall c \in \mathcal{Y}_i\}$ be its data quantity vector, with $\tilde{m}_{i,c}^t$ as the sample count of category $c \in \mathcal{Y}_i$.

**Optimization Problem**: In real-time FCL, client data arrives in small batches per FL round, resulting in less or even no overlap in label and feature distributions among some clients during that FL round. Therefore, we refer to the definitions of OCL(Gunasekara et al., 2023) and Task-Free CL (TFCL)(Wang et al., 2025), set client-specific task sequences where tasks change across FL rounds for all clients. Consider a $T$-round (i.e, $T$-tasks) FL process with a set $\boldsymbol{C} = \{C_i|i = 1, \dots, n\}$ consisting of $n$ clients, each client $C_i$ has a unique task sequence $\boldsymbol{\mathcal{T}}_i = \{\mathcal{T}_i^t|i = 1, \dots, T\}$, where each task $\mathcal{T}_i^t$ in $t$-th FL round associates to a data distribution $\mathcal{P}_i^t = P(\mathcal{X}_i^t, \mathcal{Y}_i^t)$ and a local dataset $D_i^t = \{(x_j, y_j)|\forall j \in [m_i^t]\}$ with $m_i^t$ task-ID-free samples. Similar to TFCL, in real-time FCL, each client's local datasets remain mutually exclusive across FL rounds, despite possible distribution overlaps (i.e, $D_i^{t_1} \cap D_i^{t_2} = \emptyset, \forall t_1, t_2 \in [T], t_1 \neq t_2, \forall i \in [n]$), and each $D_i^t$ is available to client $C_i$ only in $t$-th FL round, ensuring that each training sample is processed by client $C_i$ within a single FL round. With unknown task ID, while the $t$-th FL round comes, clients jointly train a global task model or $n$ personalized global task models that perform well across each client $C_i$'s all $t$ tasks' data distributions $\{\mathcal{P}_i^1, \dots, \mathcal{P}_i^t\}$. Let $\boldsymbol{\omega}_{g,i}^t$ denote the global task model received by client $C_i$, the global objective $\mathcal{F}^t$ in the $t$-th FL round can be formulated as:

$$\min_{\{\boldsymbol{\omega}_{g,i}^t|i\in[n]\}} \mathcal{F}^t(\boldsymbol{\omega}) := \frac{1}{n}\sum_{i=1}^n \mathcal{F}_i^t(\boldsymbol{\omega}_{g,i}^t); \mathcal{F}_i^t(\boldsymbol{\omega}_{g,i}^t) := \sum_{k=1}^t \mathop{E}_{(x,y)\sim\mathcal{P}_i^k}\left[l(f(\boldsymbol{\omega}_{g,i}^t; x); y)\right] \quad (1)$$

Where $\mathcal{F}_i^t$ is the local objective of client $C_i$ in $t$-th FL round (i.e., task $\mathcal{T}_i^t$), $f(\boldsymbol{\omega}_{g,i}^t; x)$ is the output of model $\boldsymbol{\omega}_{g,i}^t$ in $x$, $l(\cdot; \cdot)$ is the mission loss related to mission type (e.g., classification, or others). Note that when all clients share the same global task model $\boldsymbol{\omega}_g^t$ (i.e., $\boldsymbol{\omega}_{g,i}^t \leftarrow \boldsymbol{\omega}_g^t, \forall i \in [n]$) in each FL round $t \in [T]$, and each task span multiple FL rounds (i.e., $D_i^{t_3} = D_i^{t_4}, \mathcal{P}_i^{t_3} = \mathcal{P}_i^{t_4}, \forall i \in [n], \forall t_3, t_4$ from the same task, $t_3 \neq t_4$; $D_i^{t_5} \cap D_i^{t_6} = \emptyset, \mathcal{P}_i^{t_5} \cap \mathcal{P}_i^{t_6} \approx \emptyset, \forall i \in [n], \forall t_5, t_6$ from different tasks), objective 1 will degrade from real-time FCL to (offline) FCL.

Figure 1: The flowchart compares the generators' update mechanisms. Given a client $C_i$ with predefined categories $\{c_1, c_2, c_3, c_4\}$, the dataset $D_i^t$ for its $t$-th task contains $\{c_3, c_4\}$. In (1.a), client $C_i$ employs $G_i^{t-1}$ to create a synthetic dataset $\tilde{D}_i^{t-1}$, mixes it with $D_i^t$ to mitigate $G_i^{t-1}$'s catastrophic forgetting on $\{c_1, c_2\}$. In (1.b), client $C_i$ updates sub-generators $G_{i,c_3}^{t-1}, G_{i,c_4}^{t-1}$ on category-specific datasets $D_{i,c_3}^t, D_{i,c_4}^t \subset D_i^t$, respectively. In (1.c), client $C_i$ first employs latest task model $\boldsymbol{\omega}_i^{t,*}$ (captures features of $\{c_3, c_4\}$ from $D_i^t$) to compute the accuracies $ACC_{c_3}^t, ACC_{c_4}^t$ on $\tilde{D}_{i,c_3}, \tilde{D}_{i,c_4}$ created by $G_{i,c_3}^{t-1}, G_{i,c_4}^{t-1}$, respectively. then, with threshold $TH_G$, if $ACC_c^t < TH_G$, client $C_i$ updates $G_{i,c}^{t-1}$ on $D_{i,c}^t$ (e.g., category $c_3$), else retains $G_{i,c}^{t-1}$ (e.g., category $c_4$), reducing redundant updates.

## 4 METHODOLOGY

### 4.1 A FLEXIBLE GENERATED REPLAY ARCHITECTURE

**1. Decouple Local Generator by Category**: In (offline) FCL where tasks span multiple FL rounds, existing generative replay-based FCL methods(Yang et al., 2024) employ FL to train a large global generator which captures the data distributions of all clients' historical tasks, thereby mitigating catastrophic forgetting in the global task model (See Fig 1.a). However, in real-time FCL where tasks change per FL round, these methods face the three key challenges: Firstly, a single global model (task model or generator) struggles to rapidly adapt to all clients' local data distribution shifts(Sabah et al., 2024), resulting in convergence delays to task changes and performance degradation(Zhu et al., 2021). Secondly, a single generator requires extensive parameters to fit the complex global historical data distributions(Bubeck & Sellke, 2021), thereby elevating computational demands and delaying convergence(Gui et al., 2023). Finally, due to the inaccessibility of prior tasks' data, a single generator must self-generate replays during training to mitigate catastrophic forgetting, with more training data delaying convergence(Wang et al., 2024a). Since model's training resource needs scales with data volume under a fixed batch size(Gui et al., 2023), the computational cost of training a generator on the full dataset for one epoch equals to the sum of training it separately on each category's data for one epoch, and further equal to the sum of training the category-specific generators on their respective data for one epoch. Therefore, for each client $C_i \in C$ with local label space $\mathcal{Y}_i = \cup_{k=1}^{T} \mathcal{Y}_i^k$, we configure a smaller existing generative model for each category $c \in \mathcal{Y}_i$ as a category-specific sub-generator $G_{i,c}$ (i.e., local generator $G_i = \{G_{i,c} | \forall c \in \mathcal{Y}_i\}$). During the $t$-th FL round (task $\mathcal{T}_i^t$), let $G_i^{t-1} = \{G_{i,c}^{t-1} | \forall c \in \mathcal{Y}_i\}$ denote the local generator to be updated, client $C_i$ updates the sub-generator subset $\{G_{i,c}^{t-1} | \forall c \in \mathcal{Y}_i^t\}$ of $G_i^{t-1}$ that corresponding to task $\mathcal{T}_i^t$'s label space $\mathcal{Y}_i^t \subset \mathcal{Y}_i$, using category-specific datasets $\{D_{i,c}^t | \forall c \in \mathcal{Y}_i^t\}$ where $D_{i,c}^t = \{(x,y) | \forall (x, y = c) \in D_i^t\}$, respectively. (See Fig 1.b) At this point, the local generator effectively mitigates inter-class catastrophic forgetting, while avoiding generative replay or significant cost increase, and accelerates training.

**2. Reduce Generator Updates via Task Model**: By decoupling the generator by category, we mitigates generator's inter-class catastrophic forgetting without generative replay. However, if there is no feature drift in data with the same category across multiple tasks, updating that category's sub-generators in all tasks merely increases training burden with marginal gains. Since client updates both local task model and generator on task-specific local dataset, we propose using the updated local task model to detect feature shifts in the sub-generators to be updated, and selectively updating only those with larger feature shifts, to reduce invalid updates and further accelerate training. Taking classification mission as an example, as shown in Fig 1.c, during the $t$-th FL round (task $\mathcal{T}_i^t$), client $C_i$ first updates the global task model to obtain the local one (denoted as $\boldsymbol{\omega}_i^{t,*}$), which learns the latest features from local dataset $D_i^t$. Next, for local generator $G_i^{t-1}$'s sub-generators $\{G_{i,c}^{t-1} | \forall c \in \mathcal{Y}_i^t\}$ that to be updated, client $C_i$ uses each sub-generator $G_{i,c}^{t-1}$ to create a synthetic dataset, then employs $\boldsymbol{\omega}_i^{t,*}$ to evaluates its accuracy on class $c$ (denoted as $ACC_c^t$), if $ACC_c^t$ falls below a predefined

threshold $TH_G$, client $C_i$ updates $G_{i,c}^{t-1}$ on the category-specific dataset $D_{i,c}^t = \{(x,y)|\forall(x, y = c) \in D_i^t\}$ then get the updated sub-generator $G_{i,c}^{t,*}$, otherwise, no update is performed on $G_{i,c}^{t-1}$. Thirdly, with all updated sub-generators (denoted as the set $\{G_{i,c}^{t,*}|c \in \mathcal{Y}_i^t\}$), client $C_i$ replaces the corresponding old sub-generators in $G_i^{t-1}$, thereby obtaining the updated local generator $G_i^t$.

**3. Enhance Generative Replay via Task Model**: By using smaller, class-specific sub-generators with lower update frequency, we improve local generator's update efficiency while mitigating catastrophic forgetting, but sub-generators' limited feature-matching capability, which offering some privacy protection via underfitting, trades generation performance. Therefore, we propose using the latest task model to enhancing sub-generators' synthesis data, adapting to task model's need without improving sub-generators' generation performance. Taking classification mission as an example, given the latest task model $\boldsymbol{\omega}_i^{t,*}$ and the local generator $G_i^t = \{G_{i,c}^t|\forall c \in \mathcal{Y}_i\}$, we aim to create a synthetic dataset $\tilde{D}_i^t$ ("$\sim$" denotes "synthesis") based on a generated data quantity vector $\tilde{Y}_i^t = \{\tilde{m}_{i,c}^t|\forall c \in \mathcal{Y}_i\}$, where $\tilde{m}_{i,c}^t$ denotes the sample count for each category $c \in \mathcal{Y}_i$. For each category $c \in \mathcal{Y}_i$, we first employ the sub-generator $G_{i,c}^t$ to generate $\tilde{m}_{i,c}^t$ samples that are judged as category $c$ by $\boldsymbol{\omega}_i^*$, denoted as a small synthetic dataset $\tilde{D}_{i,c}^t = \{(\tilde{x}_j, c)|\forall j \in [\tilde{m}_{i,c}^t]\}$. Then, we merge all small synthetic datasets $\{\tilde{D}_{i,c}^t|\forall c \in \mathcal{Y}_i\}$ to form the $\tilde{D}_i^t$ (i.e., $\tilde{D}_i^t = \cup_{c \in \mathcal{Y}_i} \tilde{D}_{i,c}^t$).

**4. Adjust Local Generation Scales Adaptively**: In real-time FCL, clients' task evolution paths vary widely. Since our generator design supports category-level generation scale adjustment, to reduce replay errors from synthetic data in the task model while alleviating catastrophic forgetting, we introduce a dynamic local replay scheme that minimizes synthetic data usage: In the $t$-th FL round (task $\mathcal{T}_i^t$), for client $C_i$, define the data quantity vector $Y_i^t = \{m_{i,c}^t|\forall c \in \mathcal{Y}_i^t\}$ composed of the data

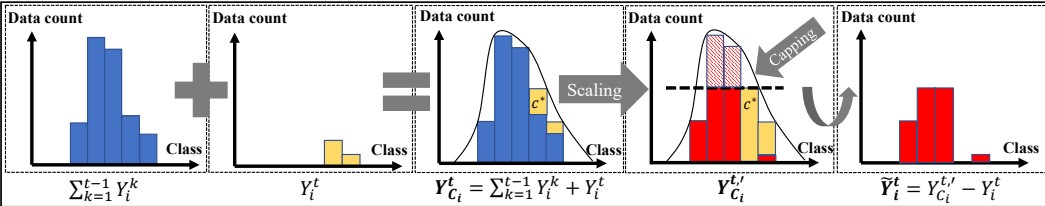

Figure 2: The flowchart of Adaptive local Generation Scale Adjustment. See text below for details.

quantities $m_{i,c}^t$ of each category $c \in \mathcal{Y}_i^t$ in local dataset $D_i^t$. As shown in Figure 2, client $C_i$ first calculates the total data quantity vector $Y_{C_i}^t = \sum_{k=1}^t Y_i^k$ ("$\sum$" represents class-wise summation) across all $t$ tasks, then proportionally scales $Y_{C_i}^t$ such that only one category of data has a quantity matching that in $Y_i^t$, denoted that category as $c^* \in \mathcal{Y}_i^t$, and then capping each category's generative scale at $m_{i,c^*}^t \in Y_i^t$, thereby obtaining the scaled data quantity vector $Y_{C_i}^{t,'}$ from $Y_{C_i}^t$. Finally, client $C_i$ computes the generated data quantity vector $\tilde{Y}_i^t = Y_{C_i}^{t,'} - Y_i^t$, using minimal synthetic data to dynamically adapt to local distribution shifts while mitigating task model's catastrophic forgetting.

### 4.2 PFEDGRP

Using the Flexible Generated Replay Architecture, we propose pFedGRP, a pFL framework via generated replay for real-time FCL. Since in real-time FCL, a single global model struggles to rapidly adapt to all clients' local data shifts(Sabah et al., 2024), we employ dual global task models: a personalized one handles client's real-time need per FL round (task), while an averaged one initializes local task model to align global feature space, with generative replay, enhanced by the personalized one, to mitigate catastrophic forgetting. We illustrate its process via the $t$-th FL round below.

**Local Training**: In $t$-th FL round (task $\mathcal{T}_i^t$), each client $C_i \in \boldsymbol{C}$ holds a local dataset $D_i^t$ and a local generator $G_i^{t-1}$ (defined in Sec 4.1.1) updated in $(t$-1)-th FL round, and receives the personalized global task model $\boldsymbol{\omega}_{g,i}^{t-1}$ and the average global task model $\boldsymbol{\omega}_g^{t-1}$ from the server. Firstly, client $C_i$ calculates the generated data volume vector $\tilde{Y}_i^t$ (Sec 4.1.4), and uses $\tilde{Y}_i^t$, $G_i^{t-1}$ and $\boldsymbol{\omega}_{g,i}^{t-1}$ to create the synthetic dataset $\tilde{D}_i^{t-1}$ (Sec 4.1.3). Then, client $C_i$ updates $\boldsymbol{\omega}_g^{t-1}$ on the merged dataset $\tilde{D}_i^{t-1} \cup D_i^t$ to minimizing the mission loss $l(\cdot; \cdot)$, while aligning the outputs of $\boldsymbol{\omega}_g^{t-1}$ and $\boldsymbol{\omega}_{g,i}^{t-1}$ via

Figure 3: The flowchart of pFedGRP for each client $C_i \in \boldsymbol{C}$ in the $t$-th FL round (task). In local training phase, client $C_i \in \boldsymbol{C}$ computes the generated data quantity vector $\tilde{Y}_i^t$ (Sec 4.1.4), uses $\tilde{Y}_i^t$, local generator $G_i^{t-1}$ and personalized global task model $\boldsymbol{\omega}_{g,i}^{t-1}$ to create a synthetic dataset $\tilde{D}_i^{t-1}$ (Sec 4.1.3), fixes $\tilde{D}_i^{t-1}$ with task-specific dataset $D_i^t$, then updates the averaged global task model $\boldsymbol{\omega}_g^{t-1}$ on it while aligning the outputs of $\boldsymbol{\omega}_g^{t-1}$ and $\boldsymbol{\omega}_{g,i}^{t-1}$ on $\tilde{D}_i^{t-1}$ via MSE loss, obtaining the local task model $\boldsymbol{\omega}_i^{t,*}$. Afterwards, client $C_i$ partially updates $G_i^{t-1}$ to $G_i^{t-1}$ with $\boldsymbol{\omega}_i^{t,*}$(Sec 4.1.2). In global aggregation phase, server sync generator caches to $G_i^t$, using $G_i^t$, $\boldsymbol{\omega}_i^{t,*}$ and $P(\tilde{Y}_{C_i}^t)$ to create a synthetic dataset $\tilde{D}_i^t$ (Sec 4.1.3), then optimizes the aggregate weights $\{w_{i,u}^t | \forall u \in [n]\}$ on $\tilde{D}_i^t$ to aggregate the personalized global task model $\boldsymbol{\omega}_{g,i}^t$, and additionally aggregate an averaged $\boldsymbol{\omega}_g^t$.

$\lambda$-weighted Mean Squared Error loss $MSE(\cdot; \cdot)$ on synthetic dataset $\tilde{D}_i^{t-1}$ to reduce feature drift. Let $\boldsymbol{\omega}_i^{t,*}$ be the updated local task model, the local objective for client $C_i$ can be formulated as:

$$\boldsymbol{\omega}_i^{t,*} \leftarrow \underset{\boldsymbol{\omega}_g^{t-1}}{argmin}\left\{ \sum_{(x,y)\in\{\tilde{D}_i^{t-1}\cup D_i^t\}} l(f(\boldsymbol{\omega}_g^{t-1}; x); y) + \lambda \sum_{x\in\tilde{D}_i^{t-1}} MSE(f(\boldsymbol{\omega}_g^{t-1}; x); f(\boldsymbol{\omega}_{g,i}^{t-1}; x)) \right\}$$

(2)

Afterwards, with the threshold $TH_G$, client $C_i$ partially update the local generator $G_i^{t-1}$ to $G_i^t$ with $\boldsymbol{\omega}_i^{t,*}$ (Sec 4.1.2), and yields the updated sub-generators $\{G_{i,c}^{t,*} | c \in \mathcal{Y}_i^t\}$. Finally, client $C_i$ normalizes the sum of all $t$ tasks' data volume vectors $\sum_{k=1}^t Y_i^k$ (denoted as $P(\tilde{Y}_{C_i}^t)$) to approximate the local label distribution, sends $\boldsymbol{\omega}_i^{t,*}$, $\{G_{i,c}^{t,*} | c \in \mathcal{Y}_i^t\}$ and $P(\tilde{Y}_{C_i}^t)$ to the server, ending local training.

**Global Aggregating**: In pFedGRP, the server keeps a local generator cache (also denoted as $G_i$) per client $C_i \in \boldsymbol{C}$ for personalized global task model aggregation. In $t$-th FL round, denote the local task models uploaded by all $n$ clients as $\{\boldsymbol{\omega}_u^{t,*} | \forall u \in [n]\}$. For each client $C_i \in \boldsymbol{C}$, server syncs the old local generator cache $G_i^{t-1}$ to $G_i^t$ via the updated sub-generators $\{G_{i,c}^{t,*} | c \in \mathcal{Y}_i^t\}$, and then uses $P(\tilde{Y}_{C_i}^t)$, $\boldsymbol{\omega}_i^{t,*}$ and $G_i^t$ to create a synthetic dataset $\tilde{D}_i^t$ (Sec 4.1.3). Let $\{w_{i,u}^t | \forall u \in [n]\}$ with $\sum_{u=1}^n w_{i,u}^t = 1$ denote the personalized aggregation weight for client $C_i$, the server updates it by minimizing the mission loss $l(\cdot; \cdot)$ of the aggregated task model $\sum_{u=1}^n w_{i,u}^t \cdot \boldsymbol{\omega}_u^{t,*}$ on $\tilde{D}_i^t$. Denoting the optimal weight as $\{w_{i,u}^{t,*} | \forall u \in [n]\}$, the personalized global objective for client $C_i$ be:

$$\{w_{i,u}^{t,*} | \forall u \in [n]\} \leftarrow \underset{\{w_{i,u}^t | \forall u \in [n]\}}{argmin} \sum_{(x,y)\in\tilde{D}_i^t} l\left( f\left( \sum_{u=1}^n w_{i,u}^t \cdot \boldsymbol{\omega}_u^{t,*}; x \right); y \right), s.t. \sum_{u=1}^n w_{i,u}^t = 1 \quad (3)$$

Finally, for client $C_i$, the server aggregates the personalized global task model $\boldsymbol{\omega}_{g,i}^t \leftarrow \sum_{u=1}^n w_{i,u}^{t,*} \cdot \boldsymbol{\omega}_u^{t,*}$ and an averaged global task model $\boldsymbol{\omega}_g^t \leftarrow \frac{1}{n}\sum_{u=1}^n \boldsymbol{\omega}_u^{t,*}$, then sends both $\boldsymbol{\omega}_{g,i}^t$ and $\boldsymbol{\omega}_g^t$ to client $C_i$. After the server completes the above process for all $n$ clients, the $t$-th FL round ends.

Appendix D presents the pFedGRP algorithm in structured pseudocode format.

# 5 EXPERIMENT

## 5.1 DATA SETTINGS, EVALUATION METRICS, DATASETS AND BASELINES

**Data Settings**: In FCL, each task spanning multiple FL rounds is assigned mutually exclusive categories while covering all their data, with $\alpha$-Dirichlet sampling commonly used to create task-specific global data heterogeneity(Yang et al., 2024). However, in real-time FCL, tasks with small data batches dynamically co-evolve with FL rounds, resulting in distinct client task paths while client

may encounter similar tasks (sharing the same data category but containing entirely different task-id-free data) in different FL rounds, the lack of intra-class data accessibility control makes $\alpha$-Dirichlet sampling ineffective in real-time FCL. Thus, we propose the following settings: Given an existing dataset with $K$ categories($N$ samples each), each client randomly divides the $K$ categories into mutually exclusive $K_{\mathcal{T}}$-sized subsets(one per task-type), then split each category's $N$ samples into $N_{\mathcal{T}}$-sized subsets for each task, forming $K_{\mathcal{T}} \cdot N_{\mathcal{T}}$-sample tasks per task-type. Ultimately, each client obtains $\lfloor K/K_{\mathcal{T}} \rfloor$ distinct task-types (varying across clients) with $\lfloor N/N_{\mathcal{T}} \rfloor$ tasks per task-type. In each FL round, each client's training data is limited to one randomly selected pending task, while the task-id-free test data accumulates across FL rounds (tasks), creating a real-time FCL process with $T = \lfloor K/(K_{\mathcal{T}} \cdot N_{\mathcal{T}}) \rfloor \cdot N$ FL rounds (tasks). More details can be found in Appendix A.1.

**Evaluation Metrics**: In real-time FCL, given the large task volume and the absence of task IDs in test data, we employ online learning's metric(Hoi et al., 2021), measuring FL method's Accuracy (Acc) and Regret (Reg) on the cumulative test data in each FL round (task). Specifically, we define a 'Centralized' baseline where clients can access all previous data per FL round (task) without global aggregation, the Regret is measured as the Accuracy gap between 'Centralized' and the FL method. Furthermore, we evaluate FL method's overall performance via Average Accuracy (AA) and Average Regret (AR) across all $T$ FL rounds (tasks). Details are provided in Appendix A.3.

**Datasets**: With above data settings and evaluation metrics, we construct three real-time FCL scenarios on six existing datasets: FashionMNIST(F-MNIST)(Xiao et al., 2017), EMNIST-Byclass(Cohen et al., 2017), CIFAR10(Krizhevsky & Hinton, 2009), CIFAR100(Krizhevsky & Hinton, 2009), plus two ImageNet(Krizhevsky et al., 2012) subsets: ImageNet-10 (random 10 categories from ILSVRC2012) and TinyImageNet-100 (top 100 categories). Details are provided in Appendix A.2.

**BaseLines**: Since existing FL and pFL methods failing to mitigate catastrophic forgetting, we select one method for each: FedAVG(McMahan et al., 2017) and pFedGraph(Ye et al., 2023). For FCL methods, we select five task-id-free methods via generative replay: FedCIL(Qi et al., 2023) and AF-FCL(Wuerkaixi et al., 2024) (Client-side generator training), as well as TARGET(Zhang et al., 2023), MFCL(Babakniya et al., 2023) and LANDER(Tran et al., 2024) (Server-side generator training via global task model). Note that most FCL methods via rehearsal or parameter isolation require task id in data, we excluded these approaches. Then we select two classic generative models with distinct principles - WGAN-GP(Gulrajani et al., 2017) and DDPM(Ho et al., 2020) - as pFedGRP's sub-generators to verify universality, denoted as pFedGRP+WGAN-GP and pFedGRP+DDPM, respectively. The details and setups of all the above methods are provided in Appendix B.1 and B.2.

## 5.2 BASELINE EXPERIMENTS

Under the above data settings, we designed three real-time FCL scenarios: the first two on F-MNIST, CIFAR-10 and ImageNet-10, the third on EMNIST, CIFAR-100 and TinyImageNet-100. All tasks comprise $K_{\mathcal{T}} = 2$ categories and $N_{\mathcal{T}} = 200$ samples per category ($N_{\mathcal{T}} = 50$ for ImageNet series). We use $T$ to abstractly denote the total number of FL rounds (tasks) across different datasets.

**Expt.1: Real-time FCL with Tasks Gradually Changing**. The scenario describes the data distribution experiencing frequent and repetitive shifts over time, mimicking real-world conditions. With $K = 10$ categories, each task-type contains $K_{\mathcal{T}} = 2$ categories, yielding random $10/2 = 5$ task-types for each client. Each client $C_i$ randomly selects two task-types (denoted as $\mathcal{T}_{i,1}, \mathcal{T}_{i,2}$) from its five to form a loop, where task execution follows the task-type sequence: $\mathcal{T}_{i,1}, \mathcal{T}_{i,2}, \mathcal{T}_{i,1}, \mathcal{T}_{i,2}$ ... After every $\lfloor T/5 \rfloor$ FL rounds, another task-type (denoted as $\mathcal{T}_{i,3}$) is chosen to replace the earlier task-type in the loop, for example, if $\mathcal{T}_{i,1}$ is replaced, the loop then consists of $\mathcal{T}_{i,2}$ and $\mathcal{T}_{i,3}$.

**Expt.2: Real-time FCL with Tasks Circulating**. The scenario involves circular data distribution changes. Each client $C_i$ is assigned five random task-types (similar to Expt.1), arranged in an ordered cycle. As FL rounds progress, task execution follows the sequence of task-types: $\mathcal{T}_{i,1}$, $\mathcal{T}_{i,2}, \mathcal{T}_{i,3}, \mathcal{T}_{i,4}, \mathcal{T}_{i,5}, \mathcal{T}_{i,1},\ldots$ repeating until all $T$ FL rounds (tasks) are completed.

**Expt.3: Real-time FCL with Extreme Data Heterogeneity**. This scenario occurs when global data heterogeneity is excessive, preventing FL methods from converging. With each task-type contains $K_{\mathcal{T}} = 2$ categories, each client is assigned random 31 task-types (62/2) on EMNIST and random 50 task-types (100/2) on both CIFAR100 and TinyImageNet-100. Each client then organizes all its task-types into a cycle and completes one iteration (consisting of 31 or 50 FL rounds (tasks)).

Table 1: Baseline Experiment Results of Expt.1 and Expt.2

| FL methods | Expt.1: Tasks Gradually Changing | | | | | | Expt.2: Tasks Circulating | | | | | |
| | F-MNIST | | CIFAR10 | | ImageNet-10 | | F-MNIST | | CIFAR10 | | ImageNet-10 | |
| | AA↑ | AR↓ | AA↑ | AR↓ | AA↑ | AR↓ | AA↑ | AR↓ | AA↑ | AR↓ | AA↑ | AR↓ |
|---|---|---|---|---|---|---|---|---|---|---|---|---|
| FedAVG | 51.39 | 37.78 | 23.79 | 36.90 | 20.23 | 34.67 | 54.68 | 32.93 | 21.06 | 35.79 | 16.94 | 31.41 |
| pFedGraph | 54.49 | 34.68 | 22.64 | 38.05 | 20.19 | 34.71 | 56.98 | 30.63 | 18.52 | 38.33 | 18.95 | 29.40 |
| FedCIL | 74.17 | 15.00 | 31.22 | 29.47 | 11.49 | 43.41 | 72.18 | 15.43 | 24.45 | 32.40 | 10.44 | 37.91 |
| AF-FCL | 73.11 | 16.06 | 29.94 | 30.75 | 20.41 | 34.49 | 70.89 | 16.72 | 21.98 | 34.87 | 26.17 | 22.18 |
| TARGET | 72.08 | 17.09 | 29.98 | 30.71 | 14.38 | 40.52 | 70.36 | 17.25 | 18.64 | 38.21 | 28.55 | 19.80 |
| MFCL | 70.85 | 18.32 | 29.14 | 31.55 | 27.54 | 27.36 | 70.11 | 17.50 | 19.70 | 37.15 | 26.15 | 22.20 |
| LANDER | 73.32 | 15.85 | 30.83 | 29.86 | 25.69 | 29.21 | 71.12 | 16.50 | 21.03 | 35.82 | 26.80 | 21.55 |
| pFedGRP+WGAN-GP | **82.80** | **6.37** | **41.94** | **18.75** | **37.17** | **17.73** | **82.34** | **5.27** | **33.53** | **23.32** | **33.68** | **14.67** |
| pFedGRP+DDPM | - | - | **52.70** | **7.99** | **49.79** | **5.11** | - | - | **46.06** | **10.79** | **43.61** | **4.74** |
| Centralized | 89.17 | 0 | 60.69 | 0 | 54.90 | 0 | 87.61 | 0 | 56.85 | 0 | 48.35 | 0 |

Table 2: Baseline Experiment Results of Expt.3

| FL methods | EMNIST-Byclass | | CIFAR100 | | TinyImageNet-100 | |
| | AA↑ | AR↓ | AA↑ | AR↓ | AA↑ | AR↓ |
|---|---|---|---|---|---|---|
| FedAVG | 5.48 | 76.67 | 2.36 | 32.11 | 2.45 | 20.72 |
| pFedGraph | 7.27 | 74.88 | 3.23 | 31.24 | 3.05 | 20.12 |
| FedCIL | 5.85 | 76.30 | 1.78 | 32.69 | 1.54 | 21.63 |
| AF-FCL | 5.31 | 76.84 | 1.74 | 32.73 | 2.33 | 20.84 |
| TARGET | 4.46 | 77.69 | 1.76 | 32.71 | 2.35 | 20.82 |
| MFCL | 4.98 | 77.17 | 1.68 | 32.79 | 2.33 | 20.84 |
| LANDER | 4.78 | 77.37 | 1.83 | 32.64 | 2.53 | 20.64 |
| pFedGRP+WGAN-GP | **51.33** | **30.82** | **9.02** | **25.45** | **8.07** | **15.10** |
| pFedGRP+DDPM | - | - | **21.85** | **12.62** | **13.91** | **9.26** |
| Centralized | 82.15 | 0 | 34.47 | 0 | 23.17 | 0 |

Table 1 and Table 2 summarize the results of three baseline experiments, while the Acc Charts in Appendix D showing all FL, pFL and FCL methods' accuracy trends across FL rounds (tasks).

In Expt.1 and Expt.2 with repeated task-types, as shown in Table 1 and the Acc Charts in Appendix E.1 and E.2, pFedGRP, equipped with category-decoupling generator and dual task models, achieves convergence with fewer FL rounds (tasks), thereby reducing regret and improving performance. In contrast, FCL methods with single generator and task model need more FL rounds (tasks) to converge and yield inferior results.

In Expt.3 with emerging new task-types continuously and non-convergence, as shown in Table 2 and the Acc Charts in Appendix E.3, pFedGRP better maintains the task model's performance on previous tasks through personalized aggregation in each FL round (task), surpassing other FCL methods.

## 5.3 ABLATION STUDIES

Our pFedGRP integrates a generative replay architecture and a pFL framework featuring dual task models. Ablation studies (AS) on FMNIST and CIFAR10 under the scenarios of Expt.1 and Expt.2 evaluated both components, with all sub-generators employing WGAN-GP. The results are presented in Table 3, while the Acc Charts in Appendix E.4 showing all methods' accuracy trends across FL rounds (tasks).

For the generated replay architecture: 'pFedGRP-AS1': Disabling the improvement of generative replay via the latest task model; 'pFedGRP-AS2': Disabling feature shift detection during the update of all sub-generators per FL round (task); 'pFedGRP-AS3': Replacing all smaller sub-generators with a dual-channel WGAN-GP model as a single larger generator, similar to other FCL methods.

For the pFL framework featuring dual task models: 'pFedGRP-ASG': Disabling the local and personalized global task models' output alignment on synthetic data (i.e., $\lambda = 0$) during local training; 'pFedGRP-ASP': Disabling initializing local task model with the averaged global task models but the personalized one; 'FedAVG-replay' and 'pFedGraph-replay': Disabling pFedGRP's pFL aggregation while integrating the generative replay architecture into FedAVG and pFedGraph.

For pFedGRP's two hyperparameters—personalized knowledge transfer weight $\lambda$ (default is 0.3) and sub-generator update threshold $TH_G$ (default is 0.25) —we analyze how they affect the task model's performance, and presents the results in Table 4 and Table 5.

Table 3: Ablation Study (AS) Results of pFedGRP's Components

| FL methods | Expt.1: Tasks Gradually Changing | | | | Expt.2: Tasks Circulating | | | |
|---|---|---|---|---|---|---|---|---|
| | F-MNIST | | CIFAR10 | | F-MNIST | | CIFAR10 | |
| | AA↑ | AR↓ | AA↑ | AR↓ | AA↑ | AR↓ | AA↑ | AR↓ |
| pFedGRP-AS1 | 81.44 | 7.68 | 37.77 | 22.91 | 80.27 | 7.34 | 28.04 | 28.81 |
| pFedGRP-AS2 | **83.67** | **5.45** | **41.36** | **19.33** | **82.64** | **4.97** | **33.54** | **23.31** |
| pFedGRP-AS3 | 82.44 | 6.68 | 29.16 | 31.53 | 79.98 | 7.63 | 20.80 | 36.05 |
| pFedGRP-ASG | 79.17 | 9.95 | 40.88 | 19.81 | 72.18 | 15.43 | 31.07 | 25.78 |
| pFedGRP-ASP | 75.82 | 13.30 | 34.08 | 26.61 | 70.36 | 17.25 | 24.69 | 32.16 |
| FedAVG-replay | 77.40 | 11.72 | 39.38 | 21.31 | 74.11 | 13.50 | 32.91 | 23.94 |
| pFedGraph-replay | 80.20 | 8.92 | 37.84 | 22.85 | 76.59 | 11.02 | 32.58 | 24.27 |
| pFedGRP+WGAN-GP | **82.80** | **6.42** | **41.94** | **18.75** | **82.34** | **5.27** | **33.53** | **23.32** |
| Centralized | 89.12 | 0 | 60.69 | 0 | 87.61 | 0 | 56.85 | 0 |

Table4: AS Results of $\lambda$ in Expt.1

| $\lambda$ | F-MNIST | | CIFAR10 | |
|---|---|---|---|---|
| | AA↑ | AR↓ | AA↑ | AR↓ |
| 0 | 79.12 | 10.00 | 40.88 | 19.80 |
| 0.1 | 81.13 | 7.99 | 41.23 | 19.46 |
| 0.2 | 81.99 | 7.13 | 42.02 | 18.67 |
| **0.3** | 82.80 | 6.32 | 41.94 | 18.77 |
| 0.4 | 82.87 | 6.24 | 40.42 | 20.28 |
| 0.5 | 83.14 | 5.97 | 38.68 | 22.01 |
| 1 | 83.22 | 5.90 | 36.96 | 23.72 |
| Centralized | 89.12 | 0 | 60.69 | 0 |

Table 5: AS Results of $TH_G$ in Expt.1

| $TH_G$ | F-MNIST | | | CIFAR-10 | | |
|---|---|---|---|---|---|---|
| | AA↑ | AR↓ | AUC↓ | AA↑ | AR↓ | AUC↓ |
| 0.1 | 81.35 | 7.77 | 10.0 | 30.12 | 30.57 | 10.0 |
| 0.2 | 82.24 | 6.88 | 16.4 | 35.10 | 25.59 | 24.2 |
| **0.25** | 82.80 | 6.32 | 22.6 | 41.94 | 18.66 | 34.4 |
| 0.3 | 83.27 | 5.85 | 26.7 | 42.13 | 18.56 | 37.5 |
| 0.4 | 83.56 | 5.56 | 38.3 | 42.45 | 18.24 | 97.8 |
| 0.5 | 83.61 | 5.51 | 73.5 | 42.32 | 18.37 | 162.4 |
| Centralized | 89.12 | 0 | - | 60.69 | 0 | - |

As shown in Table 3 and Appendix E.4: pFedGRP-AS1 underperforms pFedGRP in all FL scenarios, showing that using task model enhanced data generation is efficient; pFedGRP-AS2 shows marginal performance gain over pFedGRP, suggesting limited necessity for updating all sub-generators per FL rounds; pFedGRP-AS3 performs worst with highist computational costs, showcasing the limitations of using a single large global model; pFedGRP-ASG underperforms pFedGRP in all FL scenarios, show that personalized knowledge transfer can alleviate task model forgetting; pFedGRP-ASP lags behind pFedGRP/pFedGRP-ASG in mid-late FL stages with more categories, showing global task model initialization boosts generalization; Both FedAVG-replay and pFedGraph-replay underperform pFedGRP in the early FL stages, showing the superior efficiency of pFedGRP's pFL aggregation. As shown in Table 4, the optimal $\lambda$ correlates directly with task model performance. As shown in Table 5 where 'AUC' denotes the Average Update Count of each clients' all sub-generators across $T$ FL rounds, a lower $TH_G$ also reduces sub-generators' updates, degrading pFedGRP's performance, while a higher $TH_G$ traps pFedGRP at the sub-generators' limited generative performance, causing redundant updates.

### 5.4 ADDITIONAL EXPERIMENTAL RESULTS AND EXTENDED RESEARCH

Due to space limitations, Appendix C.1 details the generator parameters and computational costs of pFedGRP and other FCL methods, and quantifies the extra overhead from generator training. The additional ablation studies in Appendix C.2 further analyze the performance differences across generative replay schemes. Moreover, based on the setup of Expt.1, Appendix C.3 evaluates the performance variations of FL methods by adjusting the degree of cross-task data distribution shifts, and Appendix C.4 examines the effects of client volume and participation rate on FL methods.

## 6 LIMITATIONS AND FUTURE WORK

Due to space limitations, the discussion of limitations and future work is provided in Appendix F.

## 7 CONCLUSION

In this paper, we extend generative replay-based FCL to real-time FCL where clients encounter new tasks with small-batch data per FL round. To address slow convergence of the global models in existing FCL methods, we propose a flexible generative replay architecture that splits the larger single generator into category-specific smaller sub-generators, utilizes the task model to reduce the sub-generators update frequency for efficiency, and combines task-model-enhanced generative replay with adaptive local generation scale adjustment to improve the task model's catastrophic forgetting alleviation. Afterwards, we proposed pFedGRP, a generative replay-based method, achieving personalized global aggregation for clients' real-time needs per FL round (task), and enabling local personalized knowledge transfer to alleviate the task model's catastrophic forgetting, thereby improving task model's performance while reducing regrets. Experimental results show that pFedGRP outperforms other generative replay-based FCL methods across various real-time FCL scenarios.

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

# A  DATA SETTINGS, DATASETS AND EVALUATION METRICS

## A.1  DETAILS OF DATA SETTINGS

We use existing datasets to construct the real-time FCL scenarios. In our setting, the time interval between the server twice sending the task model to the clients constitutes a FL round. During each FL round, every client executes a specific task pertaining to one of its task-types (As detailed in Section 5.1). Specifically, for every client, each task-type contains multiple specific tasks that share the same categories but have different actual data. Each specific task contains training data and test data, where the training data is only accessible to the client during the FL round when executing that specific task, while the test data will be used in all FL rounds after the execution of that specific task to evaluate the performance of the task model on the client side. Figure 4 shows the schematic

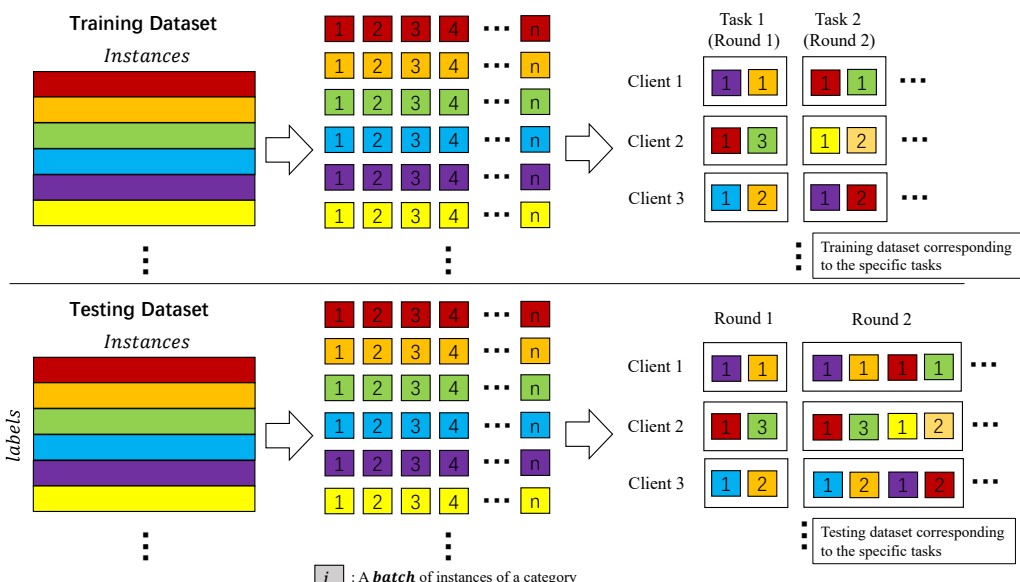

Figure 4: Schematic diagram of the partitioning of local training data and testing data. Please refer to the following text for detailed information

diagram of constructing local training and test data for each client: Each color in the figure represents one category of data, with each category containing $N$ samples. We evenly divide the training data samples of each category into $\lfloor N/N_\mathcal{T} \rfloor$ non-overlapping parts by grouping every $N_\mathcal{T}$ samples, then divide the test data samples into $N_\mathcal{T}$ equal and non-overlapping parts. In each FL round, based on the categories corresponding to the task type of the specific task executed by the client, the client selects training data parts that have not been previously accessed by the client to build the local training dataset (As shown in the upper right of Figure 4). and adds the corresponding test data parts into the local test dataset (As shown in the lower right of the Figure 4).

## A.2  DETAILS OF DATASETS

The specific details and settings of the datasets used in our experiments are as follows:

**FashionMNIST**. The FashionMNIST(F-MNIST)(Xiao et al., 2017) dataset is a 10-category clothing classification dataset (i.e., $K = 10$), with each category containing 6000 training samples and 1000 test samples (i.e., $N = 6000$), where each sample is a single-channel grayscale image of size $28 \times 28$ representing a type of clothing. In our baseline experimental setup, each client comprises 5 task types, each task type includes 2 distinct categories (i.e., $K_\mathcal{T} = 2$), and the training data allocated to each category within a specific task is 200 samples (i.e., $N_\mathcal{T} = 200$). Therefore, the total number of FL rounds in the baseline experiments for the F-MNIST dataset is calculated as $T = \lfloor 6000/(200 \times 2) \rfloor \times 10 = 150$.

**EMNIST-ByClass**. The EMNIST-ByClass(Cohen et al., 2017) dataset consists of 62 imbalanced categories of handwritten characters and numbers, containing $814255$ grayscale images sized

$28 \times 28$. Compared with the F-MNIST dataset, EMNIST-ByClass dataset contains covers categories, and its English characters incorporate both uppercase and lowercase forms, consequently increasing classification difficulty. In our baseline experimental setup, each task type includes 2 distinct categories (i.e., $K_{\mathcal{T}} = 2$), thus each client comprises $62/2 = 31$ task types, resulting in total FL rounds $T = 31$. The training data and test data allocated to each category within a specific task consist of 200 samples (i.e., $N_{\mathcal{T}} = 200$) and 100 samples, respectively.

**CIFAR10**. The CIFAR10(Krizhevsky & Hinton, 2009) dataset is a real image classification dataset consisting of 10 categories of $32 \times 32$ color RGB images (i.e., $K = 10$), each category containing 5000 training images (i.e., $N = 5000$) and 1000 test images. Compared with the MNIST series dataset, CIFAR10 contains objects in the real world which have not only have a lot of noise but also different proportions and features, making data classification more difficult. Our experimental setup on the CIFAR10 dataset is the same as that on the MNIST dataset, and the total number of FL rounds is $T = \lfloor 5000/(200 \times 2) \rfloor \times 10 = 120$.

**CIFAR100**. The CIFAR100(Krizhevsky & Hinton, 2009) dataset is a real image classification dataset consisting of 20 super categories, each super category comprises 5 categories and contains of $32 \times 32$ color RGB images, with the total number of categories being 100 (i.e., $K = 100$). Each category has 500 training images and 100 test images. Compared with the CIFAR10 dataset, the CIFAR100 dataset has a larger number of categories, and the images of each category within the same super category are more similar which increases the difficulty of classification. In our baseline experimental setup, each task type includes 2 distinct categories (i.e., $K_{\mathcal{T}} = 2$), therefore each client comprises $100/2 = 50$ task types, resulting in a total of $T = 50$ FL rounds. The training data and test data allocated to each category within a specific task consist of 200 samples (i.e., $N_{\mathcal{T}} = 200$) and 100 samples, respectively.

**ImageNet-10**. The ILSVRC2012 is a dataset used in the ImageNet(Krizhevsky et al., 2012) Large Scale Visual Recognition Challenge in 2012, consisting of approximately 1.4 million training images, 50000 validation images, and 100000 test images, covering 1000 object categories, with each category's training set containing 1300 samples. We randomly selected 10 categories of training sets from ILSVRC2012 to form the ImageNet-10 dataset (i.e., $K = 10$), and divided each category's training set into 1000 training samples (i.e., $N = 1000$) and 300 test samples in sequence. In our baseline experimental setup, each client comprises 5 task types, each task type includes 2 distinct categories (i.e., $K_{\mathcal{T}} = 2$), and the training data allocated to each category within a specific task is 50 samples (i.e., $N_{\mathcal{T}} = 50$). Therefore, the total number of FL rounds in the baseline experiments for the ImageNet-10 dataset is calculated as $T = \lfloor 1000/(50 \times 2) \rfloor \times 10 = 100$.

**TinyImageNet-100**. TinyImageNet is a subset of the ImageNet(Krizhevsky et al., 2012) dataset, containing 200 categories with 500 training images, 50 validation images, and 50 test images per category. We construct the TinyImageNet-100 dataset by selecting data from the first 100 categories in TinyImageNet. In our baseline experimental setup, each task type includes 2 distinct categories (i.e., $K_{\mathcal{T}} = 2$), thus each client comprises $100/2 = 50$ task types, resulting in total FL rounds $T = 50$. The training data and test data allocated to each category within a specific task consist of 50 samples (i.e., $N_{\mathcal{T}} = 50$) and 50 samples, respectively.

### A.3 DETAILS OF EVALUATION METRICS

Under the experimental setup above, we evaluate the performance of each FL method based on the following proposed metrics: Accuracy(Acc), Average Accuracy (AA) and Average Regret (AR). Let the client set be denoted as $C$ and the total number of FL rounds as $T$; the definitions of these metrics are as follows:

**Accuracy**. After global aggregation in each FL round $t$, we evaluate the performance of the global task models on all test data corresponding to previous $t$ tasks for each client $C_i \in C$(i.e., accuracy, denoted as $a_i^t$), and then calculate the Acc of the $t$-th FL round based on the weighted average of the total number of training data encountered by each client $C_i$ (denoted as $n_i^t$):

$$Acc^t = \frac{1}{\sum_{C_i \in C} n_i^t} \sum_{C_i \in C} n_i^t \cdot a_i^t \qquad (4)$$

The Acc can indicate the comprehensive performance of the global task model obtained in FL round $t$ across all previously encountered tasks.

**Average Accuracy**. This metric uses the mean Acc value across all $T$ FL rounds to indicate the overall performance of each FL method throughout the entire FL process, that is:

$$AA = \frac{1}{T} \sum_{t=1}^{T} Acc^t \tag{5}$$

AA can reduce evaluation errors caused by variations in task difficulty and better assess the performance stability of different methods throughout the FL process.

**Average Regret**. the Average Regret (AR) is the difference between the performance of the client when it can access real data of previous tasks and when it cannot access such data. Let the Acc of the Centralized method in the $t$-th FL round be $Acc_{Centralized}^t$, the Average Regret (AR) of each method is:

$$AR = \frac{1}{T} \sum_{t=1}^{T} (Acc^t - Acc_{Centralized}^t) \tag{6}$$

AR can evaluate the extent to which the performance of FL methods declines as the number of tasks increases, with a smaller value indicating better memory stability of the FL method.

# B  DETAILS OF BASELINE METHODS AND EXPERIMENTAL SETUP

## B.1  DETAILS OF BASELINE METHODS

We compare pFedGRP with following one FL methods, one pFL methods and five task-id-free generative replay-based FCL methods, and establish the performance of local task models - where clients can access real data from previous tasks - as the theoretical boundary, referred to as the "Centralized" methods. The FL methods and pFL methods lack the capability to retain information related to historical tasks, while the task-id-free FCL methods can mitigate catastrophic forgetting to some extent. In ablation studies, we further integrate FL and pFL methods with our generated replay framework.

**FedAVG**: FedAVG(McMahan et al., 2017) is a representative federated learning approach, it constructs the global model through weighted aggregation of client-uploaded parameters, where the aggregation weights correspond to the proportion of each client's local training data volume.

**pFedGraph**: pFedGraph(Ye et al., 2023) is a relatively new personalized federated learning method. It proposes constructing a personalized collaboration graph on the server based on the cosine differences between local task models, thereby enabling personalized aggregation of the global task model for individual clients to balance local utility and collaborative benefits. Additionally, it employs cosine similarity constraints during local training to mitigate model deviation.

**FedCIL**: FedCIL(Qi et al., 2023) is a newer Federated Continual Learning (FCL) method based on the ACGAN framework, which integrates both the task model and generator into a unified ACGAN architecture. During local training phases, FedCIL leverages data generated by both the global AC-GAN model and previous local ACGAN models to mitigate catastrophic forgetting in local ACGAN models through model distillation and label alignment techniques. In the global aggregation phase, the server first performs model averaging on local ACGAN models to obtain the global ACGAN model, then fine-tunes this global model using data generated by each local ACGAN model.

**AF-FCL**: AF-FCL(Wuerkaixi et al., 2024) is a relatively new FCL method based on local sample-free generated replay and distillation. It designs a local distillation mechanism based on partial feature forgetting. During the client-side local training phase, to achieve the dual objectives of distilling data features for the task model and obtaining a higher-performing generator, AF-FCL alternately trains the local generator and task model using both real data and data replayed by the

global generator. On the server side, it employs an averaging method to aggregate both the task models and generators

**TARGET**: TARGET(Zhang et al., 2023) is a relatively new FCL method based on global feature replay. On the server side, it trains a global generator using aggregated batch normalization (BN) layer features from the global task model and an untrained task model. This enables the global generator to synthesize data that can be effectively classified by the task model. On the client side, it alleviates catastrophic forgetting in the task model by leveraging replayed data generated by the global generator.

**MFCL**: MFCL(Babakniya et al., 2023) is a relatively new FCL method based on global sample-free generated replay and distillation. On the server side, it proposes a training scheme that leverages the aggregated global task model to train a global generator for producing high-quality synthetic data. On the client side, during local training, the knowledge from the global task model is transferred to the local task model through distillation using the data generated by the global generator.

**LANDER**: LANDER(Tran et al., 2024) is an improved version of TARGET, it employs a pre-trained language model to produce label text embeddings that serve as anchors in the global generator's feature space. Subsequently, it leverages an aggregated global task model to train a unified global generator on the server side, which corresponds to all previously encountered tasks.

**FedAVG-replay**: FedAVG-replay extends the FedAVG algorithm by integrating pFedGRP's generated replay architecture into local training phases.

**pFedGraph-replay**: pFedGraph-replay extends the pFedGraph algorithm by integrating pFedGRP's generated replay architecture into local training phases.

**Centralized**: The Centralized method does not perform global aggregation, and each client can access the real data encountered in previous FL rounds during local training.

## B.2 DETAILS OF EXPERIMENTAL SETUP

For the task model, we choose ResNet20(He et al., 2016) for all FL methods except FedCIL. The local training epochs are uniformly set to 20 (with AF-FCL set to 100 to alternate training its local generator), and we employ SGD as the optimizer with a learning rate of 0.01, momentum of 0.9, and weight decay of 0.01. In pFedGRP, the personalized knowledge transfer weight $\lambda$ is set to 0.3. For FedCIL, the task model is the ACGAN model, which follows its default dataset-specific settings with local training epochs set to 400.

For the sub-generators of pFedGRP, we set the WGAN-GP(Gulrajani et al., 2017) model and DDPM(Ho et al., 2020) model to perform updates using their respective default loss functions and settings, with an update threshold $TH_G = 0.25$. For MNIST-series datasets, we deploy the WGAN-GP model with 16 channels as sub-generator, which undergoes 200 training epochs when updates are required. For CIFAR-series datasets, we implement two types of existing generative models as sub-generators: the 64-channel WGAN-GP and the default-configured DDPM, which are trained for 500 and 4000 epochs per update cycle, respectively. For ImageNet-series datasets, we implement two types of existing generative models as sub-generators: the 128-channel WGAN-GP and the default-configured DDPM, which are trained for 1000 and 6000 epochs per update cycle, respectively.

For the generators of other FCL methods, we follow their default training settings. Specifically: For the AF-FCL method, the local generator undergoes 100 epochs of training during the local training phase, which alternates with the task model training. For the FedCIL method, the local generator adopts the ACGAN model and completes 400 training epochs during the local training phase per FL round. For the TARGET, MFCL and LANDER methods, their global generators are trained on the server side. After aggregating the global task model, the server performs 100 training epochs on the global generator per FL round.

In the fine-tuning setting for global aggregation on the server side, our pFedGRP performs 20 epochs of personalized aggregation weight optimization for each client, employing optimizer settings consistent with local training. The FedCIL method performs 100 epochs of model distillation on the global ACGAN model using default configurations. Other FL methods do not involve a fine-tuning phase during the global aggregation process.

# C  ADDITIONAL EXPERIMENTAL RESULTS AND EXTENDED RESEARCH

## C.1  DETAILS OF GENERATORS' COMPUTATION AND COMMUNICATION COST

Firstly, we have compiled in Table 6 both the computational consumption per data sample (i.e., FLOPs) and the model parameters of all models utilized in our experimental study. Then, we cal-

Table 6: FLOPs and Parameters of all models used in the experiments

| Models | MNIST series dataset | | CIFAR series dataset | | ImageNet series dataset | |
|---|---|---|---|---|---|---|
| | FLOPs | Parameter | FLOPs | Parameter | FLOPs | Parameter |
| ResNet-20 | 29.05M | 701.18K | 35.66M | 701.466K | 35.77M | 702.23K |
| ResNet-20 (AF-FCL) | 29.09M | 734.20K | 35.69M | 734.490K | 35.80M | 735.26K |
| WGAN-GP (pFedGRP) | 7.19M | 186.27K | 94.54M | 1732.22K | 338.40M | 6101.25K |
| DDPM (pFedGRP) | - | - | 4061.68M | 167726.40K | 16239.97M | 167726.40K |
| ACGAN (FedCIL) | 241.10M | 3951.69K | 957.47M | 14719.17K | 1157.97M | 18197.96K |
| Flow (AF_FCL) | 46.49M | 4663.81K | 176.87M | 17715.71K | 688.74M | 68985.34K |
| Generator (TARGET) | 89.21M | 1834.31K | 117.70M | 2328.90K | 470.81M | 8644.93K |
| GEN(MFCL) | 93.76M | 6500.87K | 123.64M | 8423.94K | 507.77M | 8571.78K |
| Generator (LANDER) | 3173.19M | 130826.60K | 3173.19M | 130826.60K | 43504.28M | 425186.42K |
| WGAN-GP (pFedGRP-AS3) | 24.87M | 536.38K | 356.852M | 6085.888K | - | - |

culate the average training iterations of the generator for pFedGRP and each FCL method in the Baseline experiment:

In the 150 FL rounds (tasks) on the F-MNIST dataset, each pFedGRP client trained the WGAN-GP sub-generator an average of 24.7 times, each time using only half of the local data (due to the local data containing two types), while other FCL methods trained the global generator 150 times.

In the 120 FL rounds (tasks) on the Cifar10 dataset, each pFedGRP client trained the WGAN-GP sub-generator an average of 36.3 times and the DDPM sub-generator an average of 10 times, each time using only half of the local data, while other FCL methods trained the global generator 120 times.

In the 100 FL rounds (tasks) on the ImageNet-10 dataset, the client trained the WGAN-GP sub-generator an average of 32.6 times and the DDPM sub-generator an average of 10 times, each time using only half of the local data, while other FCL methods trained the global generator 100 times.

We have listed in Tables 7, Tables 8, and Tables 9 the average additional computational cost per data by the training generator for pFedGRP and various FCL methods.

Table 7: The average computational cost of training generator on F-MNIST datasets

| FL methods | Local computational cost | | | | Global computational cost | | | Avg Acc | |
|---|---|---|---|---|---|---|---|---|---|
| | Total times | Local epoch | Model FLOPs | **Total FLOPs per data** | Global epoch | Model FLOPs | **Total FLOPs per data** | Expt.1 | Expt.2 |
| FedCIL | 150 | 400 | 241.1M | 14466B | 100 | 241.1M | 3617B | 74.17 | 72.18 |
| AF-FCL | 150 | 100 | 46.5M | 1395B | - | - | 0 | 73.11 | 70.89 |
| TARGET | 150 | - | - | 0 | 100 | 89.21M | 1338B | 72.08 | 70.36 |
| MFCL | 150 | - | - | 0 | 100 | 93.76M | 1406B | 70.85 | 70.11 |
| LANDER | 150 | - | - | 0 | 100 | 3173.19M | 47609B | 73.32 | 71.12 |
| pFedGRP-AS3 | 150 | 200 | 24.8M | 744B | - | - | 0 | 82.44 | 79.98 |
| pFedGRP+ WGAN-GP | 24.7*0.5 | 200 | 7.2M | 35.568B | - | - | 0 | **82.80** | **82.34** |

Table 8: The average computational cost of training generator on Cifar10 datasets

| FL methods | Local computational cost | | | | Global computational cost | | | Avg Acc | |
|---|---|---|---|---|---|---|---|---|---|
| | Total times | Local epoch | Model FLOPs | Total FLOPs per data | Global epoch | Model FLOPs | Total FLOPs per data | Expt.1 | Expt.2 |
| FedCIL | 120 | 400 | 957.5M | 45960B | 100 | 957.5M | 11490B | 31.22 | 24.45 |
| AF-FCL | 120 | 100 | 176.9M | 2123B | - | - | 0 | 29.94 | 21.98 |
| TARGET | 120 | - | - | 0 | 100 | 117.70M | 1412B | 29.98 | 18.64 |
| MFCL | 120 | - | - | 0 | 100 | 123.64M | 1484B | 29.14 | 19.70 |
| LANDER | 120 | - | - | 0 | 100 | 3173.19M | 38078B | 30.83 | 21.03 |
| pFedGRP-AS3 | 120 | 500 | 356.9M | 21414B | - | - | 0 | 29.16 | 20.80 |
| pFedGRP+ WGAN-GP | 36.3*0.5 | 500 | 94.5M | 858B | - | - | 0 | **41.94** | **33.53** |
| pFedGRP+ DDPM | 10*0.5 | 4000 | 4061.7M | 81234B | - | - | | **52.70** | **46.06** |

Table 9: The average computational cost of training generator on ImageNet-10 datasets

| FL methods | Local computational cost | | | | Global computational cost | | | Avg Acc | |
|---|---|---|---|---|---|---|---|---|---|
| | Total times | Local epoch | Model FLOPs | Total FLOPs per data | Global epoch | Model FLOPs | Total FLOPs per data | Expt.1 | Expt.2 |
| FedCIL | 100 | 400 | 1158.0M | 46320B | 100 | 1158.0M | 11580B | 11.49 | 10.44 |
| AF-FCL | 100 | 100 | 688.7M | 6887B | - | - | 0 | 20.41 | 26.17 |
| TARGET | 100 | - | - | 0 | 100 | 470.8M | 4708B | 14.38 | 28.55 |
| MFCL | 100 | - | - | 0 | 100 | 507.8M | 5078B | 27.54 | 26.15 |
| LANDER | 100 | - | - | 0 | 100 | 43504.3M | 435043B | 25.69 | 26.80 |
| pFedGRP+ WGAN-GP | 32.6*0.5 | 1000 | 338.4M | 5516B | - | - | 0 | **37.17** | **33.68** |
| pFedGRP+ DDPM | 10*0.5 | 6000 | 16240.0M | 487200B | - | - | 0 | **49.79** | **43.61** |

As shown in the tables above, pFedGRP outperforms other FCL methods with lower additional training consumption when WGAN-GP is employed as the sub-generator. In contrast, using DDPM as the sub-generator results in higher training consumption but delivers even stronger performance.

## C.2 ABLATION STUDIES ON PFEDGRP'S GENERATORS' PERFORMANCE

In addition to the ablation experiments provided in the main text, we calculated the FID(Heusel et al., 2017) values of the generated replay schemes used by various methods in the ablation study. The lower the value, the better the performance of the generated replay. The results are shown in Table 10 below:

Table 10: FID Values for various Generated Replay Schemes on pFedGRP

| Generated Replay Scheme | F-MNIST | CIFAR10 | ImageNet-10 |
|---|---|---|---|
| | Fid↓ | Fid↓ | Fid↓ |
| A single Double-Channel WGAN-GP | 301.390 | 707.879 | 974.345 |
| Only WGAN-GP as sub-generator | 187.622 | 436.116 | 407.823 |
| WGAN-GP as sub-generator + ResNet20 | 165.552 | 390.213 | 376.938 |
| Only DDPM as sub-generator | - | 77.035 | 181.153 |
| DDPM as sub-generator + ResNet20 | - | 65.284 | 149.714 |

It can be seen that as the complexity of data increases, the generated replay effect of the auxiliary model with category decoupling gradually becomes much better than that of a single larger auxiliary model. On this basis, using the information contained in the task model can further enhance the generated replay performance of the auxiliary model. However, due to the underfitting update strategy adopted by pFedGRP to reduce the update frequency, significant distribution differences persist between the generated data and the real data, while the resulting distribution noise enhances the privacy protection capability of pFedGRP.

## C.3 EXTEND EXPERIMENTS UNDER DIFFERENT DATA HETEROGENEITY

Under the baseline experimental setting of Expt.1, we further investigate how the progressively strengthened correlations between tasks affect the performance of various FL methods. Since the number of duplicate categories between adjacent tasks for each client in the baseline experimental setting is 0, we increased this value to 2, 4, and 6 (i.e., each task type consequently contains 4, 6, and 8 categories respectively) while maintaining the real data quantity per category at 200 (50 for the ImageNet-10 dataset). As the number of duplicate categories between adjacent tasks grows, data heterogeneity progressively decreases. The performance of pFedGRP and other baseline methods under these experimental configurations is documented in Table 11, Table 12 and Table 13.

Table 11: Extend Experiment Results on F-MNIST and the setting of Expt.1

| FL methods | The number of duplicate categories between adjacent tasks on each client | | | | | | | |
| | 0 | | 2 | | 4 | | 6 | |
| | AA↑ | AR↓ | AA↑ | AR↓ | AA↑ | AR↓ | AA↑ | AR↓ |
|---|---|---|---|---|---|---|---|---|
| FedAVG | 51.39 | 37.78 | 75.61 | 12.71 | 83.70 | 5.150 | 84.61 | 3.27 |
| pFedGraph | 54.49 | 34.68 | 74.18 | 14.14 | 81.98 | 6.870 | 81.43 | 6.44 |
| FedCIL | 74.17 | 15.00 | 83.25 | 5.08 | 87.35 | 1.50 | 84.59 | 3.30 |
| AF-FCL | 73.11 | 16.06 | 83.15 | 5.18 | 87.79 | 1.06 | 85.41 | 2.45 |
| TARGET | 72.08 | 17.09 | 81.47 | 6.85 | 86.44 | 2.42 | 83.94 | 3.95 |
| MFCL | 70.85 | 18.32 | 82.41 | 5.91 | 86.61 | 2.24 | 84.48 | 3.41 |
| LANDER | 73.32 | 15.85 | 82.93 | 5.39 | 87.04 | 1.81 | 84.76 | 3.12 |
| pFedGRP+ WGAN-GP | **82.80** | **6.37** | **84.86** | **3.46** | **87.81** | **1.04** | **86.41** | **1.47** |
| Centralized | 89.17 | 0 | 88.32 | 0 | 88.85 | 0 | 87.88 | 0 |

Table 12: Extend Experiment Results on Cifar10 and the setting of Expt.1

| FL methods | The number of duplicate categories between adjacent tasks on each client | | | | | | | |
| | 0 | | 2 | | 4 | | 6 | |
| | AA↑ | AR↓ | AA↑ | AR↓ | AA↑ | AR↓ | AA↑ | AR↓ |
|---|---|---|---|---|---|---|---|---|
| FedAVG | 23.79 | 36.90 | **50.97** | **13.24** | **58.05** | **9.18** | 63.30 | 5.421 |
| pFedGraph | 22.64 | 38.05 | 50.15 | 14.05 | **56.70** | **10.53** | 62.37 | 6.351 |
| FedCIL | 31.22 | 29.47 | 39.57 | 24.63 | 44.59 | 22.64 | 44.57 | 24.15 |
| AF-FCL | 29.94 | 30.75 | 44.93 | 19.28 | 47.24 | 19.99 | 49.63 | 19.09 |
| TARGET | 29.98 | 30.71 | 42.35 | 21.85 | 45.37 | 21.85 | 48.42 | 20.30 |
| MFCL | 29.14 | 31.55 | 45.92 | 18.29 | 46.21 | 21.01 | 46.50 | 22.22 |
| LANDER | 30.83 | 29.86 | 45.73 | 18.48 | 45.96 | 19.53 | 48.55 | 20.17 |
| pFedGRP+ WGAN-GP | **41.94** | **18.75** | 48.60 | 15.60 | 47.70 | 19.53 | 50.76 | 17.96 |
| pFedGRP+ DDPM | **52.70** | **7.99** | **55.43** | **8.77** | 56.11 | 11.12 | 56.53 | 12.19 |
| Centralized | 60.69 | 0 | 64.21 | 0 | 67.23 | 0 | 68.72 | 0 |

Table 13: Extend Experiment Results on ImageNet-10 and the setting of Expt.1

| FL methods | The number of duplicate categories between adjacent tasks on each client | | | | | | | |
| | 0 | | 2 | | 4 | | 6 | |
| | AA↑ | AR↓ | AA↑ | AR↓ | AA↑ | AR↓ | AA↑ | AR↓ |
|---|---|---|---|---|---|---|---|---|
| FedAVG | 20.23 | 34.67 | 33.77 | 24.53 | 40.87 | 18.53 | 45.22 | 15.03 |
| pFedGraph | 20.19 | 34.71 | 34.77 | 23.53 | 38.45 | 20.95 | 44.39 | 15.92 |
| FedCIL | 11.49 | 43.41 | 9.72 | 48.57 | 9.84 | 49.55 | 9.59 | 50.73 |
| AF-FCL | 20.41 | 34.49 | 37.62 | 20.68 | 43.83 | 15.56 | 44.73 | 15.59 |
| TARGET | 14.38 | 40.52 | 19.08 | 39.22 | 37.94 | 21.46 | 40.37 | 19.95 |
| MFCL | 27.54 | 27.36 | 38.27 | 20.03 | 44.35 | 15.04 | **45.30** | **15.02** |
| LANDER | 25.69 | 29.21 | 37.40 | 20.90 | 45.48 | 13.91 | 44.96 | 15.36 |
| pFedGRP+ WGAN-GP | **37.17** | **17.73** | **44.27** | **14.08** | **45.63** | **13.76** | 45.28 | 15.04 |
| pFedGRP+ DDPM | **49.79** | **5.11** | **52.83** | **5.47** | **52.36** | **7.03** | **51.52** | **8.80** |
| Centralized | 54.90 | 0 | 58.30 | 0 | 59.39 | 0 | 60.32 | 0 |

As evidenced by the above tables, all FL methods exhibit a trend of performance improvement with decreasing data heterogeneity. However, on datasets like CIFAR10 which feature relatively complex data distributions but narrower performance bottlenecks for task models, the generation error caused by the generator will have a significant impact on the task model, resulting in five FCL methods and the pFedGRP method underperforming compared to FL and pFL methods in scenarios with low data heterogeneity. Nevertheless, by employing multiple strategies to reduce generative-replay errors, the pFedGRP method consistently outperforms all FCL methods. For the more complex ImageNet-10 dataset, where even task models encounter performance bottlenecks, the impact of generative replay errors becomes less significant compared to that of data scarcity. This comparative advantage causes the FedCIL method based on the ACGAN model to completely fail to converge, while enabling the pFedGRP method using DDPM as a sub-generator to achieve optimal performance.

## C.4 Extend Experiments under Different Client States

Under the baseline experimental setting of Expt.1, we further investigated the performance impact of different total client counts and varying client participation ratios on various FL methods on the Cifar10 dataset, in order to explore the robustness of each method.

For varying counts of clients, we examined scenarios with 5, 10, 20, and 30 clients while maintaining identical experimental conditions, where the baseline configuration uses 10 clients by default. Since the complexity of the global data distribution remains constant, increasing the number of clients allows the global model to better capture the overall data features, thereby accelerating model convergence. The performance of pFedGRP and other baseline methods under varying client counts is presented in Table 14.

Table 14: Extend Experiment Results on Cifar10 and the setting of Expt.1

| FL methods | The number of total client count | | | | | | | |
| | 5 | | 10 | | 20 | | 30 | |
| | AA↑ | AR↓ | AA↑ | AR↓ | AA↑ | AR↓ | AA↑ | AR↓ |
|---|---|---|---|---|---|---|---|---|
| FedAVG | 21.24 | 39.45 | 23.79 | 36.90 | 25.14 | 35.55 | 26.40 | 34.28 |
| pFedGraph | 21.13 | 39.55 | 22.64 | 38.05 | 24.86 | 35.83 | 25.93 | 34.76 |
| FedCIL | 30.24 | 30.45 | 31.22 | 29.47 | 34.75 | 25.94 | 35.68 | 25.01 |
| AF-FCL | 28.65 | 32.03 | 29.94 | 30.75 | 34.13 | 26.56 | 35.18 | 25.50 |
| TARGET | 28.11 | 32.58 | 29.98 | 30.71 | 33.96 | 26.73 | 35.42 | 25.27 |
| MFCL | 27.96 | 32.72 | 29.14 | 31.55 | 33.01 | 27.67 | 35.27 | 25.42 |
| LANDER | 28.34 | 32.35 | 30.83 | 29.86 | 35.62 | 25.07 | 35.85 | 24.84 |
| pFedGRP+ WGAN-GP | **40.71** | **19.98** | **41.94** | **18.75** | **42.14** | **18.55** | **42.18** | **18.51** |
| Centralized | 60.69 | 0 | 60.69 | 0 | 60.69 | 0 | 60.69 | 0 |

As evidenced by the above tables, the performance of all FL methods improves as the number of clients increases. The performance improvement of the FCL method, which requires global information to construct the generative model, shows a significant positive correlation with the number of clients. Compared to other methods, the performance bottleneck of pFedGRP mainly originates from the performance limitations of its generative model, resulting in lower sensitivity to client scale expansion. Nevertheless, in horizontal comparisons, pFedGRP still demonstrates superior performance, with its comprehensive evaluation metrics consistently outperforming other FL methods.

For varying client participation ratios, within the baseline setting comprising 10 clients total, we examine scenarios where 0, 1, 2, or 4 clients are randomly missing in each FL round. The default baseline setting reflects 0 missing clients per FL round. Increasing the number of missing clients per round will reduce the convergence speed of the task model, further challenging the robustness of FL methods. Table 15 records the performance of pFedGRP and other FL methods with varying client participation ratios.

Table 15: Extend Experiment Results on Cifar10 and the setting of Expt.1

| FL methods | The number of random missing client in each FL round | | | | | | | |
| | 0 | | 1 | | 2 | | 4 | |
| | AA↑ | AR↓ | AA↑ | AR↓ | AA↑ | AR↓ | AA↑ | AR↓ |
|---|---|---|---|---|---|---|---|---|
| FedAVG | 23.79 | 36.90 | 23.47 | 37.22 | 22.94 | 37.74 | 21.60 | 39.08 |
| pFedGraph | 22.64 | 38.05 | 22.55 | 38.14 | 22.37 | 38.32 | 21.98 | 38.71 |
| FedCIL | 31.22 | 29.47 | 31.18 | 29.51 | 30.98 | 29.71 | 30.50 | 30.18 |
| AF-FCL | 29.94 | 30.75 | 29.80 | 30.89 | 29.58 | 31.11 | 29.17 | 31.52 |
| TARGET | 29.98 | 30.71 | 29.03 | 31.65 | 28.86 | 31.82 | 28.30 | 32.39 |
| MFCL | 29.14 | 31.55 | 28.86 | 31.83 | 28.65 | 32.04 | 28.03 | 32.65 |
| LANDER | 30.83 | 29.86 | 28.54 | 32.15 | 28.57 | 32.12 | 28.19 | 32.50 |
| pFedGRP+ WGAN-GP | **41.94** | **18.75** | **41.92** | **18.70** | **41.80** | **18.88** | **41.55** | **19.14** |
| Centralized | 60.69 | 0 | 60.69 | 0 | 60.69 | 0 | 60.69 | 0 |

As evidenced by the above table, the performance of all FL methods deteriorates as the number of randomly missing clients per FL round increases. Similar to the findings from the previous experiment, compared with other methods, pFedGRP exhibits reduced sensitivity to client absence and demonstrates greater robustness.

# D  PSEUDOCODE FOR PFEDGRP

---

**Algorithm 1** : pFedGRP

---

1: **Input:** Client set $\boldsymbol{C} = \{C_i | i = 1, \ldots, n\}$ with $n$ clients; Global Task model $\boldsymbol{\omega}$; Local Generators $G_i^0 = \{G_{i,c}^0 | \forall c \in \mathcal{Y}_i\}$ for each client $C_i$ with local label space $\mathcal{Y}_i$.

2: **Output:** Personalized global task models $\{\boldsymbol{\omega}_{g,i}^t | \forall i \in [n]\}$ of $n$ clients in each FL round $t \in \{1, \ldots, T\}$ (i.e., the $t$-th task for each client).

3: Server random initializes $\boldsymbol{\omega}$, takes it as global task model $\boldsymbol{\omega}_g^0$ and $n$ personalized global task models $\{\boldsymbol{\omega}_{g,i}^0 | \forall i \in [n]\}$, then sends $\boldsymbol{\omega}_{g,i}^0, \boldsymbol{\omega}_g^0$ to each client $C_i \in \boldsymbol{C}$.

4: **for** each FL round $t = 1, \ldots, T$ (i.e., the $t$-th task) **do**

5:     // Client local training

6:     **for** each client $C_i \in \boldsymbol{C}$ in parallel **do**

7:         Client $C_i$ receives the personalized global task model $\boldsymbol{\omega}_{g,i}^{t-1}$ and the averaged global task model $\boldsymbol{\omega}_g^{t-1}$ from server.

8:         Client $C_i$ obtains the local dataset $D_i^t$ of task $\mathcal{T}_i^t$ along with its data quality vector $Y_i^t$ and label space $\mathcal{Y}_i^t$.

9:         Client $C_i$ computes the generated data quality vector $\tilde{Y}_i^t$ (as described in Sec 4.1.4).

10:        Client $C_i$ utilizes the local generator $G_i^{t-1}$, $\tilde{Y}_i^t$ and $\boldsymbol{\omega}_{g,i}^{t-1}$ to create the synthetic dataset $\tilde{D}_i^{t-1}$ (as described in Sec 4.1.3).

11:        Client $C_i$ updates $\boldsymbol{\omega}_g^{t-1}$ on $\left\{\tilde{D}_i^{t-1} \cup D_i^t\right\}$ by optimizing Formula (2), then obtains the local task model $\boldsymbol{\omega}_i^{t,*}$.

12:        Client $C_i$ partially updates the sub-generators in $G_i^{t-1}$ on $D_i^t$ with $\boldsymbol{\omega}_i^{t,*}$ (as described in Sec 4.1.2), then obtains $G_i^t$ and the updated sub-generators $\{G_{i,c}^{t,*} | c \in \mathcal{Y}_i^t\}$.

13:        Client $C_i$ computes the approximate local label distribution $P(\tilde{Y}_{C_i}^t) = Norm(\sum_{k=1}^t Y_i^k)$.

14:        Client $C_i$ sends $\boldsymbol{\omega}_i^{t,*}$, $\{G_{i,c}^{t,*} | c \in \mathcal{Y}_i^t\}$ and $P(\tilde{Y}_{C_i}^t)$ to the server.

15:    **end for**

16:    // Server aggregating

17:    **for** each cilent $C_i \in \boldsymbol{C}$ **do**

18:        Server synchronizes the local generator cache $G_i^{t-1}$ to $G_i^t$ with $\{G_{i,c}^{t,*} | c \in \mathcal{Y}_i^t\}$.

19:        Server utilizes $G_i^t$, $P(\tilde{Y}_{C_i}^t)$ and $\boldsymbol{\omega}_i^{t,*}$ to create the synthetic dataset $\tilde{D}_i^t$ (as described in Sec 4.1.3).

20:        Server optimizes Formula (3) on $\tilde{D}_i^t$ then obtains the optimal personalized aggregated weights $\{w_{i,u}^{t,*} | \forall u \in [n]\}$.

21:        Server aggregates the personalized global task model $\boldsymbol{\omega}_{g,i}^t \leftarrow \sum_{u=1}^n w_{i,u}^{t,*} \cdot \boldsymbol{\omega}_u^{t,*}$.

22:        Server aggregates the averaged global task model $\boldsymbol{\omega}_g^t \leftarrow \frac{1}{n} \sum_{i=1}^n \boldsymbol{\omega}_u^{t,*}$.

23:        Server sends both $\boldsymbol{\omega}_{g,i}^t, \boldsymbol{\omega}_g^t$ to client $C_i$.

24:    **end for**

25: **end for**

---

# E    ACC VARIATION CHARTS FOR EXPERIMENTS

## E.1    ACC VARIATION CHARTS FOR EXPT.1

In Expt.1, the gray vertical lines in the charts indicate the FL rounds (i.e., occurring every T/5 FL rounds) at which the task-types in each client's task-loop change.

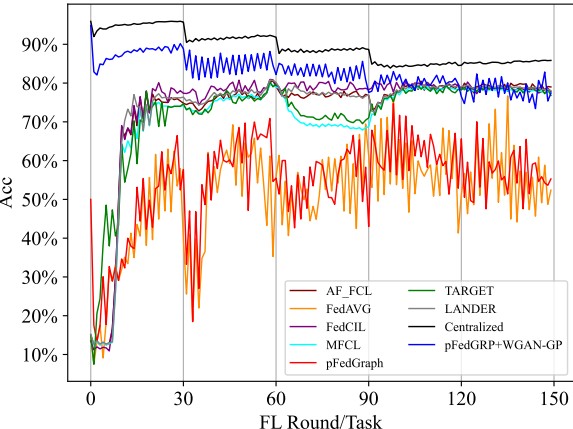

Figure 5: Acc Variation Chart on F-MNIST for Expt.1

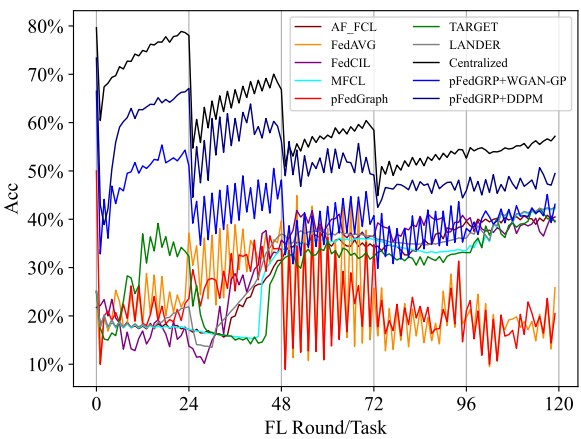

Figure 6: Acc Variation Chart on Cifar10 for Expt.1

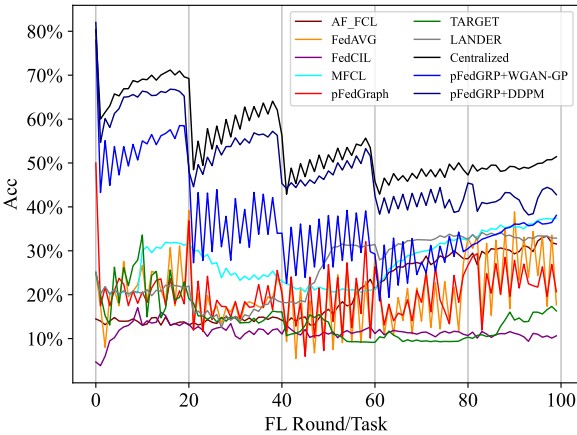

Figure 7: Acc Variation Chart on ImageNet-10 for Expt.1

### E.2 ACC VARIATION CHARTS FOR EXPT.2

In Expt.2, the gray vertical lines in the charts indicate the first FL round (i.e.,occurring every five FL rounds) of the new task-type cycle on each client.

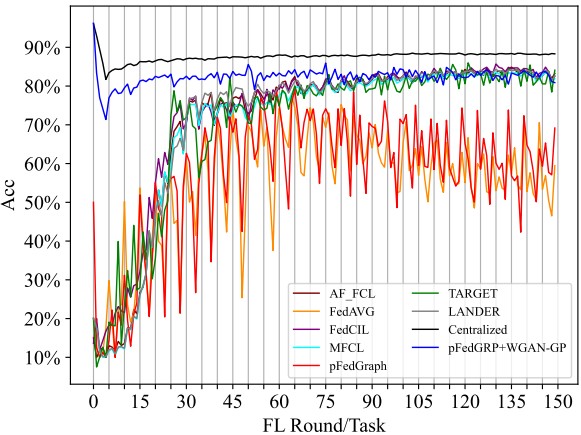

Figure 8: Acc Variation Chart on F-MNIST for Expt.2

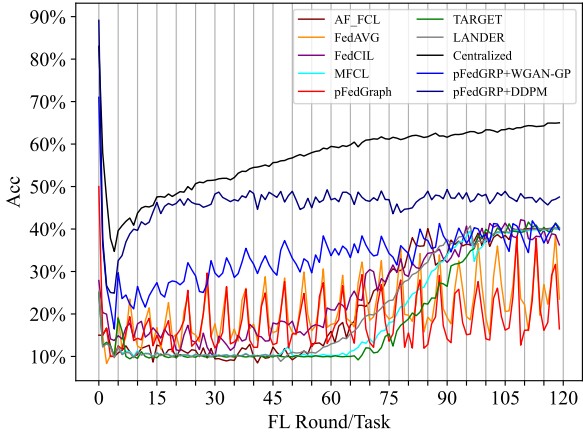

Figure 9: Acc Variation Chart on Cifar10 for Expt.2

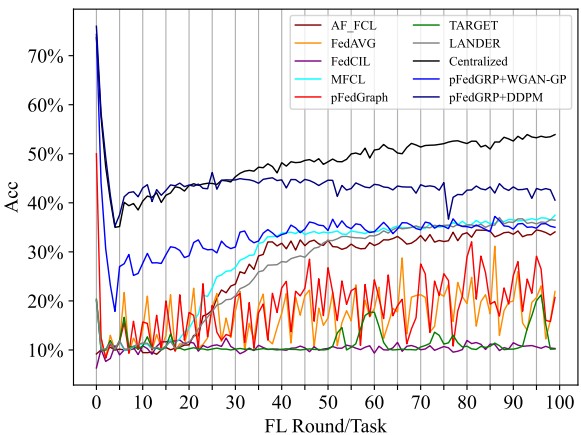

Figure 10: Acc Variation Chart on ImageNet-10 for Expt.2

## E.3 ACC VARIATION CHARTS FOR EXPT.3

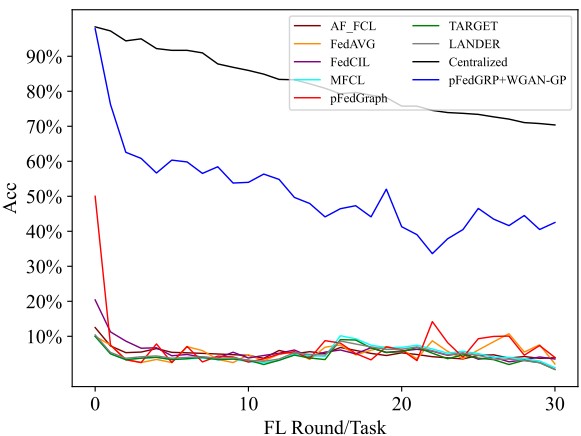

Figure 11: Acc Variation Chart on EMNIST-ByClass for Expt.3

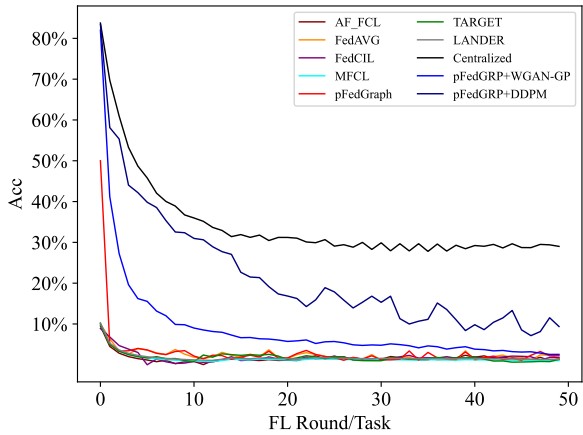

Figure 12: Acc Variation Chart on Cifar100 for Expt.3

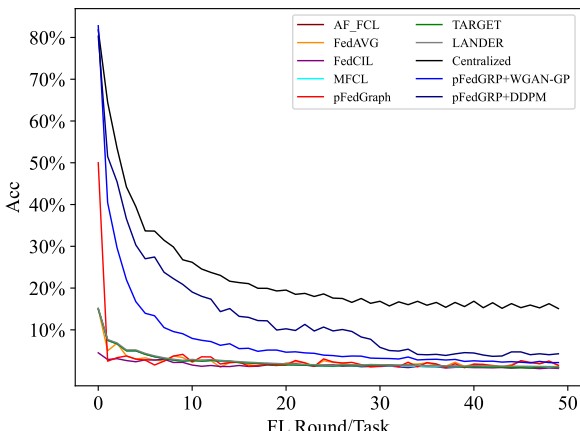

Figure 13: Acc Variation Chart on TinyImageNet-100 for Expt.3

## E.4 ACC VARIATION CHARTS FOR ABLATION STUDIES (AS)

In Expt.1, the gray vertical lines in the charts indicate the FL rounds (i.e., occurring every T/5 FL rounds) at which the task-types in each client's task-loop change.

In Expt.2, the gray vertical lines in the charts indicate the first FL round (i.e.,occurring every five FL rounds) of the new task-type cycle on each client.

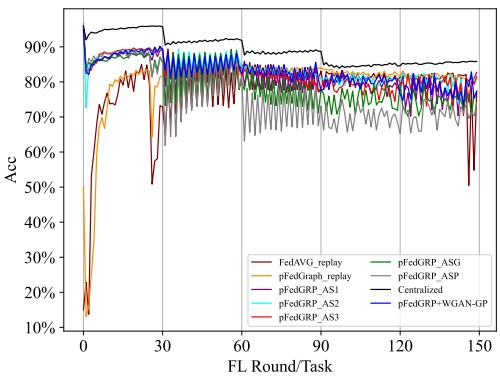

Figure 14: Acc Variation Chart on F-MNIST for AS in Expt.1

Figure 15: Acc Variation Chart on Cifar10 for AS in Expt.1

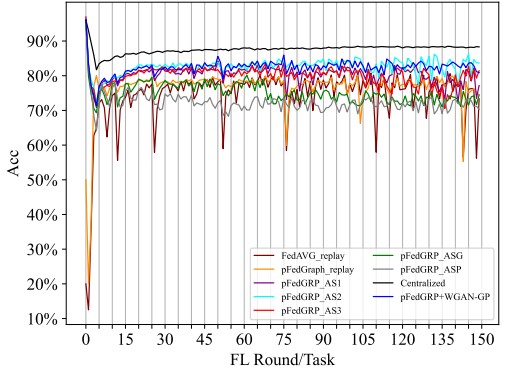

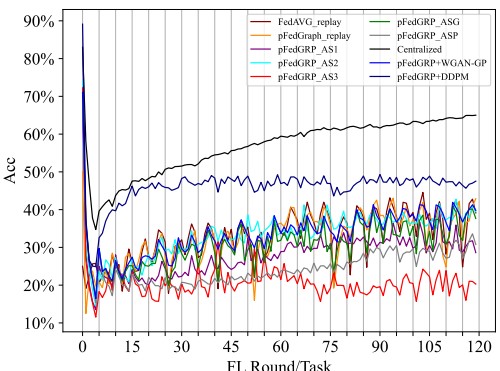

Figure 16: Acc Variation Chart on F-MNIST for AS in Expt.2

Figure 17: Acc Variation Chart on Cifar10 for AS in Expt.2

# F  LIMITATIONS AND FUTURE WORK

## F.1  LIMITATIONS

**Potential High Storage Space Requirement**: To mitigate the larger single generator's training burden caused by catastrophic forgetting through self-replay and to accelerate its updates for supporting real-time FCL, we propose a category-decoupled generative framework that assigns dedicated smaller sub-generators to each data category. Although the sub-generators utilize smaller existing generation models, they still require significant client storage space in highly diverse category scenarios. In summary, our generative replay architecture enhances training efficiency and reduces communication overhead (by transmitting only updated sub-generators) at the expense of storage space, thereby better supporting real-time FCL.

**Potential Privacy Risks**: In order to enable simultaneous real-time personalized aggregation while mitigating catastrophic forgetting, unlike the FCL methods that train the generator on the server based on the global task model, pFedGRP requires clients to train the generator locally and synchronize it with the server. Although our strategy—using smaller existing generative models as sub-generators and reducing their update frequency to accelerate generator updates—can indirectly increase generative noise, servers could still infer private information from the generated data, and this limitation is inherent to all FCL methods that train generators on the client side. To address this issue, using the existing generative model with integrated differential privacy technology as sub-generators could be a potential solution. It is worth noting that pFedGRP confines potential privacy leakage to the server side by restricting the sub-generator's transmission to a unidirectional flow (client-to-server only), thereby preventing leakage to other clients.

## F.2  FUTURE WORK

For Future Work, there are two possible directions for expansion:

Firstly, unlike existing FCL methods that rely on generative replay, pFedGRP imposes no architectural constraints on the generator or task model. This flexibility has the potential to be extended to diverse applications beyond image classification, such as regression mission and reinforcement learning scenarios.

Secondly, to address the limitation of multiple sub-generators potentially consuming excessive client storage space, more efficient approaches can be explored to balance training speed and storage requirements.

