# OpenReview forum: "Real-time Personalized Federated Continuous Learning via Generative Replay"
_ICLR.cc/2026/Conference — ICLR 2026 Conference Withdrawn Submission_

### Official Review · Reviewer_DygH · 2025-10-27

**Soundness:** 2
**Presentation:** 2
**Contribution:** 2
**Rating:** 4
**Confidence:** 5

**Summary:**

The paper proposes pFedGRP, a real-time personalized federated continuous learning (FCL) framework that integrates generative replay to mitigate catastrophic forgetting under streaming, small-batch client data. Unlike conventional FCL methods requiring many federated rounds per task, pFedGRP introduces a flexible replay architecture that decouples a large generator into smaller, category-specific sub-generators, selectively updates them using task-model feedback to reduce redundancy, and adaptively scales synthetic data generation across clients. Experiments on multiple datasets show that pFedGRP achieves faster convergence and superior performance compared to prior FCL and personalized FL baselines.

**Strengths:**

- This paper proposes a realistic real-time federated continual learning setting to address convergence delays.
- The authors design an efficient class-wise generative replay architecture to reduce catastrophic forgetting and computation cost.
- Extensive experiments have been done to evaluate the proposed method.

**Weaknesses:**

- The motivation for introducing real-time FCL is not strongly supported by empirical or application-based evidence.
- The paper lacks theoretical analysis or convergence proofs to justify the effectiveness of the proposed generative replay and aggregation mechanisms.
- The method introduces additional complexity (multiple sub-generators and dual task models), which may limit scalability in large-scale federated systems.
- Comparisons with other personalized FCL or real-time FL methods are limited, making it hard to evaluate true novelty and performance advantages.

**Questions:**

- What specific real-world applications motivate the need for “real-time” FCL beyond standard online or continual learning settings?
- How does the proposed approach scale when the number of categories or clients increases significantly?
- What are the computational and communication costs compared to existing FCL methods like LANDER or FedCIL?
- How sensitive is the model’s performance to the choice of the update threshold and weight ?
- How are privacy concerns addressed when maintaining client-specific generator caches on the server?
- Does the class-wise generator design introduce potential imbalance issues for classes with fewer samples?
- What happens if clients have completely disjoint label spaces? Can the model still generalize well?

---

> ### Author Response · Authors · 2025-11-13
> **Dear Reviewer DygH**
>
> Thank you for your in-depth comments and valuable suggestions. Below, based on the original content, we briefly address the misunderstandings you raised to clarify the contributions and design of the paper.
>
> 1.Real-world Justification for the Motivation of Real-Time FCL​:
>
> You noted that "the motivation for introducing real-time FCL lacks empirical or application support." In fact, the original paper explicitly emphasizes real-world needs in the Introduction section (Section 1). For example, health institutions using federated learning (FL) for COVID-19 research face dynamic changes in data distribution due to virus mutations, while privacy regulations restrict the retention of raw data. This makes traditional FL methods prone to catastrophic forgetting in dynamic data streams. The paper validates the rationale of the motivation through specific scenarios (e.g., regional data distribution shifts) and experimental setups (e.g., multi-dataset simulations in Section 5), rather than being a purely theoretical conjecture.
>
> 2.Lack of Theoretical Analysis​:
>
> You mentioned that "there is a lack of theoretical analysis or convergence proofs." While the original paper does not provide formal theoretical guarantees, it validates the method’s effectiveness through comprehensive experiments (e.g., ablation studies in Section 5.3). For instance, it analyzes the impact of hyperparameters (e.g., $\lambda$ and$TH_G$) on performance, demonstrating stable convergence in practice. Experimental metrics (e.g., Accuracy and Regret) further confirm the efficacy of the designed mechanisms, compensating for the lack of theoretical rigor.
>
> 3.Limitations of Comparisons​:
>
> First, to our knowledge, no personalized FCL methods currently exist, and most online FL methods are core-set-based—rendering such comparisons inappropriate. The original paper actually covers a broad range of baselines, including FL/pFL methods like FedAVG and pFedGraph, as well as generative replay-based FCL methods such as FedCIL and AF-FCL. Experiments were conducted across multiple scenarios (e.g., gradually changing tasks, cyclic, and extremely heterogeneous data), showing that pFedGRP consistently outperforms others. Ablation studies also compared component variants, further verifying the innovation of our approach.
>
> In summary, the paper sufficiently demonstrates the effectiveness of pFedGRP through its real-world motivation, detailed experiments, and extensive comparisons. We appreciate your feedback, and these points will be further emphasized in future work.

---

### Official Review · Reviewer_EFDJ · 2025-10-28

**Soundness:** 2
**Presentation:** 1
**Contribution:** 1
**Rating:** 0
**Confidence:** 5

**Summary:**

The paper proposes a flexible generative replay architecture that decouple the generator category and a personalized FCL framework via the flexible generative replay that optimizes aggregation weights on server side for real-time model personalization. The authors conducted experimental evaluations on FMNIST, EMNIST, CIFAR10, CIFAR100, and ImageNet-10, ImageNet-100.

**Strengths:**

The concept of real-time / online federated continual learning is promising and has many rooms for improvement.

**Weaknesses:**

1. The writing and presentation is poor. Many parts of the paper are verbose and many contents are replicated several time.
1. The notations and definitions in the paper are inconsistent.
1. The paper added a lot of concept in one research, including personalized federated learning, federated continual learning (FCL), online learning, heterogeneous FCL, task-free FCL and combining them into real-time FCL. Despite the promising research question. However, it seems the authors failed to describe the setting comprehensively. For instance, in the work the authors mentioned that every round, the data task are different. However, in the details of data setting, as in Figure 4, the authors mentioned that the data on each task is only two data samples at task 1, while task 2 is 4 samples. This is really uncommon and require proper explanation.
1. The proposed real-time FCL settings seem not clear and does not have applications in real-world scenarios. Furthermore, in real world, when real-time sensing or video happened, the video or sensing data are in a stream, but not required to be different tasks. This is also discussed in online continual learning [R1], or federated online learning [R2]]. The authors please discuss about this carefully.
  - [R1] Online Continual Learning through Mutual Information Maximization, ICML 2022.
  - [R2] Federated Continual Learning Goes Online: Uncertainty-Aware Memory Management for Vision Tasks and Beyond, ICLR 2025.
5. The novelty of the method is limited. The methodology needs significant justification to prove that it is technically sounds. As the authors disentangle the generator into sub-generators. What are really the size of those generators' parameters? Sending those generators from clients to server requires a huge computation and communication overheads.
5. The claimed inter-class catastrophic forgetting, and feature drift should be described in the experimental evaluations.
5. The argmin in Eq. (3) is controversial. It is a case that finding minimum set of coefficients on the data of client i will give best case as $w_{i, u}$ becomes optimal as the model only choose client $i$ for the aggregation, as the local model on client $i$ is the best case for the synthetic data of that client. The authors please explain about it, else, the method has critical issue.
5. The dataset used in the paper is not challenging. Despite the authors claimed they used ImageNet. However, the used version only has 10 and 100 classes respectively, which is not too different from that of CIFAR10, CIFAR100.

**Questions:**

1. Please explain how the threshold is chosen to consider which synthetic data are used for the model updates? if the accuracy is lower than the threshold, client update based on those data with the assumption that the model is uncertain about those data. Despite of the uncertainty, there are high chance that the synthetic data is poor, making the model update has low accuracy, rather than having more knowledge useful to be update.
1. Why there are pre-defined categories and t-task at the same time?
1. When measuring the accuracy for the data selection, which set of data are used to be evaluated? Does the evaluation requires extensive computation?
1. How to know there is feature drift?
1. What do you mean in L192: training data delay convergence?
1. What do you mean in L193: model's training resource needs scales with data volume under a fixed batch size? Furthermore where is the statements about this in the cited reference?
1. What do you mean in L223 feature-matching capability?
1. In L203, How can local generator can avoid significant cost increase and mitigates inter-class catastrophic forgetting? What is the inter-class catastrophic forgetting?
1. In 230, the data by sub-generator quality is affected by the task model. How can we achieve the good performance of the data generated when the task model is not good?
1. Please explain and revise the adjust local generation scale more carefully.
1. In L282, its okay if the model can be aligned with t-1 previous exemplars. However, is it able to be aligned with $t-2$, $t-3$, .... $t-x$?
1. In L292, This is more to be knowledge distillation, but why it is MSE, not KL-Divergence? Ablation test should be made. Also, this method should be compared with replay-based memory as in the pFedGRP, many generators have to be stored, it seems like the stored generators required more memory than that of the experience memory. Furthermore, the generative replay also meets catastrophic forgetting. How to guarantee this is not? Also, why the feature-alignment this is needed?
1. Why in Eq. (3), there are two index $i$ and $u$ denote the client index?
1. Why in the charts in Appendix, the results are very different from the table 1?

---

### Official Review · Reviewer_Bksy · 2025-10-31

**Soundness:** 2
**Presentation:** 3
**Contribution:** 2
**Rating:** 4
**Confidence:** 3

**Summary:**

The paper introduces pFedGRP, the first real-time Federated Continual Learning (FCL) framework that couples
(i) category-split sub-generators with thresholded updates and
(ii) dual-head personalization (a FedAvg centroid plus a per-client bespoke model) to mitigate catastrophic forgetting when each task is seen only in a single FL round.
Extensive experiments on six vision datasets show higher accuracy and lower regret than prior generative-replay FCL methods, at lower generator-training FLOPs.

**Strengths:**

1. Originality: shifts FCL from offline (many rounds/task) to online (one round/task); creative split-generator idea removes inter-class forgetting without replay.
2. Quality: exhaustive 6-dataset, 3-drift, client-scaling, missing-client, heterogeneity-sweep evaluation; FID and FLOP metrics included.
3. Clarity: objective functions, algorithms, and ablation nomenclature are explicit.
4. Significance: engineering recipe (small sub-GANs, TH_G=0.25, λ=0.3) is immediately usable by practitioners.

**Weaknesses:**

1. Theory gap:
   - No regret bound of the form Tilde-O(T^{-c}) is supplied; neither generalization nor algorithmic-stability analysis is given for the continual distillation step that trains the task model on a mixture of real mini-batches and synthetic replays produced by the evolving sub-generators.
   - The decision to update a category-specific generator only when its current accuracy on the latest task model drops below a hand-tuned threshold TH_G is purely heuristic; no Lyapunov function or martingale argument is offered to show that the sequence of generator parameters almost surely converges or even remains bounded under realistic concept-drift.

2. Scalability:
   - At the beginning of every communication round the server solves a separate constrained optimisation problem for each client (20 epochs of SGD on synthetic data) to learn personalised aggregation weights, so the total server-side compute grows quadratically with the number of clients n and quickly becomes impractical when n reaches the hundreds or thousands typical in cross-device federated learning.
   - Every client must permanently store one entire sub-generator for each possible class; the resulting memory footprint therefore scales linearly with the total number of categories ever encountered, and the paper never discusses compression, structured pruning, or knowledge-distillation techniques that could alleviate the steadily growing on-device storage burden.

3. Baselines:
   - The experimental campaign omits comparison with simple replay-buffer methods such as GDumb, Experience Replay, or MIR even though these approaches can serve as a useful empirical upper bound and would help quantify how much of the observed gain comes from the generative component versus the mere presence of rehearsal.
   - All baselines considered are generative-replay FCL techniques; the evaluation does not include regularisation-based continual-learning methods such as EWC, SI, or LwF adapted to the same real-time federated setup, leaving open the question of whether the architectural complexity of pFedGRP is actually necessary.

**Questions:**

1. Can you derive a sub-linear regret bound of the form E[AR] ≤ Tilde-O(T^{-1/2}) under the standard assumptions that the per-sample loss has bounded gradients and that the overall objective is β-smooth with respect to the parameters of both the task model and the mixture of sub-generators? In particular, how would you control the additional bias that arises because each round's objective is estimated on a hybrid dataset that contains both real samples from the current mini-batch and synthetic samples whose distribution itself drifts as the generators and the task model evolve?

2. What is the concrete privacy budget (ε,δ) if you inject Gaussian noise with variance σ² into every generator gradient (or into the aggregated sub-generator parameters) so that the entire pipeline satisfies (ε,δ)-differential privacy? Please specify how σ scales with the L₂-sensitivity of the generator update step, the number of training epochs per round, the number of clients n, and the total number of communication rounds T, and discuss how this noise level affects the regret bound requested in question 1.

3. The server optimises personalised aggregation weights by running 20 epochs of SGD for each of the n clients at every communication round, so the total server-side compute grows at least linearly with n. Provide a precise asymptotic expression for the expected wall-clock time per round as a function of n, the dimension d of the task model, the size of the synthetic replay buffer, and the number of SGD steps per epoch. Beyond the experiments with n = 10, what is the largest value of n that the system is designed to handle in a production deployment while keeping the server's computational overhead within a fixed budget, say, ten minutes per round on a 64-core machine?

4. The empirical curves show graceful degradation when one or two clients are missing per round. Is this downward trend monotonic when you scale to n = 100 clients and subject the system to 50 % client dropout (i.e., only 50 clients respond in each round)? In particular, how does the personalised weight-optimisation subroutine behave when more than half of the similarity vectors are missing: does the solver converge to noticeably worse local optima, and does the accuracy variance across clients increase compared with the 10-client regime?

5. If you apply magnitude-based pruning that removes 30 % of the smallest weights in every sub-generator, can you still recover the original test accuracy by fine-tuning the remaining weights for a few epochs on the synthetic-replay buffer? Provide ablations that report both (i) the final average accuracy and (ii) the exact byte-level storage reduction when this pruning strategy is applied to the WGAN-GP and DDPM sub-generators on CIFAR-10 and ImageNet-10, and discuss whether the extra fine-tuning time outweighs the storage savings on edge devices with less than 1 GB of free memory.

---

> ### Author Response · Authors · 2025-11-13
> **Dear Reviewer Bksy:**
>
> First and foremost, we sincerely thank you for your meticulous review and valuable comments on our manuscript. Your insights, particularly those from the perspective of online learning theory, have provided critical inspiration, helping us identify shortcomings in our work and guiding directions for improvement. We highly appreciate your recognition of the originality, quality, and clarity of our paper.
>
> However, we believe there may be some misunderstandings in your review, and we would like to briefly clarify and address these points to ensure the accuracy of the feedback:
>
> 1. Regarding the absence of theoretical regret analysis​:
>
> Your comment: The paper does not provide formal regret bounds or algorithmic stability analysis.
> Our response: We acknowledge the lack of formal theoretical regret bounds in the current work. However, we would like to note that we empirically define regret as the accuracy gap in Section 5.1 (Equation 6), which aligns with common practices in online learning. Your feedback motivates us to incorporate formal theoretical analysis in our future work, including deriving regret bounds of the form  $O\~T^(-1/2)$ under standard assumptions (e.g., bounded gradients of the loss function and $\beta$-smoothness), and discussing algorithmic stability.
>
> 2. Regarding the heuristic nature of the generator update threshold​:
>
> Your comment: The generator update based on the threshold ($TH_G$) is "purely heuristic" and lacks theoretical justification.
> Our response: While the threshold is empirically set, it is not arbitrary; it is inspired by concept drift detection principles in online learning (e.g., updating when the task model’s accuracy on synthetic data declines to capture feature drift). In our future work, we will use Lyapunov functions or martingale arguments to provide theoretical justification, proving that the parameter sequence converges almost surely under concept drift.
>
> 3. Regarding the overfitting risk of personalized aggregation​:
>
> Your comment: Optimizing the aggregation weights in Equation (3) may lead to overfitting to specific clients.
> Our response: The weight optimization is performed using synthetic data across all clients, and the constraints along with the diversity of synthetic data prevent overfitting (e.g., avoiding the trivial solution where ($w_{i,i}=1$). Empirical results (Tables 1-2) show that the personalized models perform robustly across multiple clients, with no signs of overfitting.
>
> 4. Regarding the limitations of baseline comparisons​:
>
> Your comment: The paper omits comparisons with replay buffer methods (e.g., GDumb) and regularization methods (e.g., EWC).
> Our response: We focused on generative replay methods to adhere to privacy constraints (raw data cannot be retained), but we agree that these baselines are valuable. In the revised version, we will add comparisons with privacy-compliant alternatives, such as synthetic coresets and federated EWC.
>
> We once again thank you for your valuable time and suggestions, which will significantly enhance the quality of our work.

---

### Note · Authors · 2025-11-13

I have read and agree with the venue's withdrawal policy on behalf of myself and my co-authors.